# Speculative Streaming: Fast LLM Inference without Auxiliary Models

## Abstract

Speculative decoding is a prominent technique to accelerate large language model inference by leveraging predictions from an auxiliary draft model. While effective, in application-specific settings, it often involves fine-tuning both draft and target models to achieve high acceptance rates. As the number of downstream tasks grows, draft models add significant complexity to inference systems. Recently several single model architectures viz. Medusa have been proposed to speculate tokens in non-autoregressive manner, however, their effectiveness is limited due to lack of dependency between speculated tokens. We introduce a novel speculative decoding method that integrates drafting within the target model by using Multi-stream attention and incorporates future token planning into supervised fine-tuning objective. To the best of our knowledge, this is the first parameter-efficient approach that scales well with an increasing number of downstream tasks while enhancing downstream metrics and achieving high acceptance rates, attributable to the interdependence among the speculated tokens. Speculative Streaming speeds up decoding by 1.9 - 3X in a diverse set of tasks, such as Summarization, Structured Queries, and Meaning Representation, while improving generation quality and using $\sim$10000X fewer extra parameters than alternative architectures, making it ideal for resource-constrained devices. Our approach can also be effectively deployed in lossless mode for generic chatbot applications where speculative instruction tuning is performed while keeping base model frozen. In such setups, we achieve 2.9 - 3.2X speedup while maintaining the integrity of the base model's output.

## 1 Introduction

Large transformers are today's preeminent tool for language modeling. The quality of these models improves as they scale (Kaplan et al., 2020), leading to the introduction of the state-of-the-art multi-billion parameter models (Brown et al., 2020; Thoppilan et al., 2022; Chowdhery et al., 2023; Touvron et al., 2023a). While these models are effective for token generation, they incur a high inference cost as the model and its transient states need to be loaded into computing memory for each subsequently generated token. This poses a challenge to the deployment of large autoregressive transformers, particularly for user-facing applications with stringent latency requirements.

Given the memory-bound nature of large language model (LLM) inference, recent work (Leviathan et al., 2023; Chen et al., 2023) proposed Speculative Decoding as an effective technique to accelerate decoding based on concepts borrowed from speculative computation (Burton, 1985) to exploit the available extra compute. The core of speculative decoding is the idea of speculating multiple candidate future tokens first, and then verifying them all in parallel. To achieve this, as shown in Figure 1a.(i), a two-model paradigm approach is used: a small auxiliary "draft" model for candidate speculation and a large "target" model for verification (Leviathan et al., 2023; Chen et al., 2023). Although effective in accelerating LLMs, speculative decoding complicates deployment. Training also becomes more demanding and complicated, as a separate draft model needs to be trained and aligned with the target model for each application. It is also not resource-efficient, requiring to host two models in memory during token prediction.

In this paper, we propose *Speculative Streaming*, a single-model speculative decoding approach that unifies speculation and verification, obviating the need for a separate draft model as shown in Figure 1a.(ii). This is accomplished by incorporating multi-stream attention into the target model to

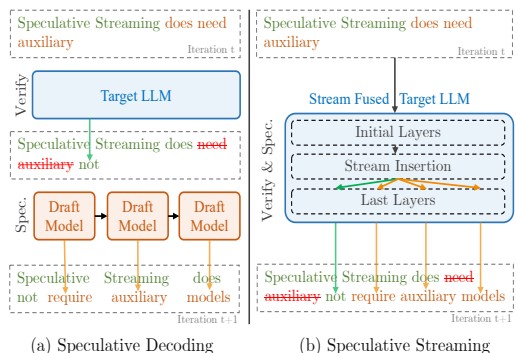

(a) Speculative Decoding

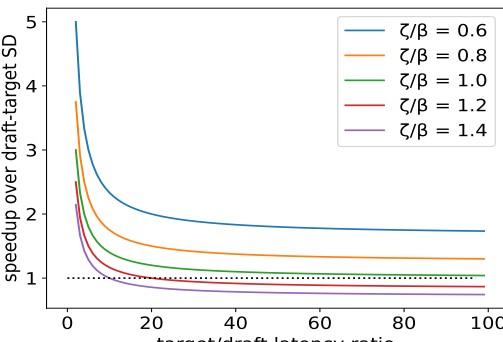

(b) Speculative Streaming

(a) (i). Speculative Decoding requires a well-aligned draft model that runs autoregressive speculation. (ii). Speculative Streaming significantly simplifies the system by performing speculation and verification concurrently, within a single stream-fused model.

(b) Theoretical speedups of Speculative Streaming compared to draft-based speculative decoding for various $\zeta/\beta$ and target-to-draft latency ratios, where $\zeta$ and $\beta$ represent the advancement per step for speculative decoding and Speculative Streaming, respectively.

Figure 1: Speculative Decoding vs Speculative Streaming

perform n-gram prediction which serves as future candidate speculation. As a result, a forward model pass can verify the previously generated tokens while simultaneously speculating on the future tokens. In theory, as token progression per forward pass approaches that of the two-stage decoding paradigm, our method consistently yields decoding speedups that surpass those achieved by the two-model approach as depicted in Figure 1b.

Our findings reveal that Speculative Streaming not only simplifies both the training and inference architectures and enhances resource efficiency, but also improves generation quality across a broad range of downstream tasks. Furthermore, it outperforms two-model speculative decoding (Leviathan et al., 2023) and single model approaches such as Medusa (Cai et al., 2023), Lookahead Decoding (Fu et al., 2023), Hydra (Ankner et al., 2024), and Eagle (Li et al., 2024) in terms of decoding speedup. The key advantages of Speculative Streaming are as follows:

– Achieves substantial decoding speedups and improves downstream performance metrics through a single, streamlined fine-tuning process leveraging multi-stream attention.
– Demonstrates resource efficiency with significantly fewer additional parameters compared to Medusa (Cai et al., 2023), Hydra (Ankner et al., 2024) and Eagle (Li et al., 2024), while still surpassing them in speedup gains.
– Simplifies deployment by removing the complexity of managing, aligning, and switching between multiple models during inference, as required by approaches like (Leviathan et al., 2023).
– Supports shared mode for application-specific scenarios, enhancing quality of responses and lossless mode for general-purpose chatbot-like settings, maintaining model's original output distribution.

## 2 RELATED WORKS

The original speculative decoding approach (Chen et al., 2023; Leviathan et al., 2023) utilizes a smaller draft model to generate a candidate sequence of tokens to be verified by the *target model*. Recent SD variants propose parallel computation along the batch axis (Sun et al., 2023b), and tree-structured batches (Miao et al., 2023; Spector & Re, 2023) to improve the acceptance rates of the guessed tokens by the target model and to further boost the performance. However, these methods encounter a common limitation: the necessity of developing an accurate and independent draft model for each downstream application. First, training such a draft model aligned with the main model is not trivial (Zhou et al., 2023). Second, hosting two different models increases the system complexity, and is more computationally and operationally expensive to maintain as number of applications grow.

Recently, single-model speculation has also been considered. In particular, inspired by (Qi et al., 2020; Stern et al., 2018), Medusa (Cai et al., 2023) extends the main model to predict future tokens in parallel by training multiple output heads. While it does not require a draft model, each Medusa head of size (*hidden_size* × *vocab_size*) requires significant nonnegotiable additional parameters which

introduce deployment challenges on resource-constrained devices. Furthermore dependency between speculated tokens is not guranteed (Ankner et al., 2024) limiting speedups. (Ankner et al., 2024) improves speculation procedure of (Cai et al., 2023) by using autoregressive draft head to introduce dependency between speculated tokens, however small size draft head tends to be sub-optimal and increasing draft head size leads to similar issues as those with (Leviathan et al., 2023; Zhou et al., 2023). (Li et al., 2024) uses a dedicated layer of target model to generate speculation, however, speedups are limited due to auto-regressive draft generation. Moreover, using a dedicated layer leads to significant parameter overhead. Lookahead decoding (Fu et al., 2023) proposes a parallel decoding method without learning new parameters. While this approach is parameter efficient, speedups are limited as speculation procedure is not learnable. We discuss further approaches related to inference efficiency in Appendix J.

## 3 METHOD

### 3.1 MOTIVATION

Existing speculative decoding techniques often enforce a strict decoupling of the training objectives between draft and target models (Leviathan et al., 2023), or between draft models and auxiliary heads (Cai et al., 2023). While this separation has been effective, we propose that these objectives are not inherently orthogonal. Instead, they can be aligned during training. Specifically, we hypothesize that, similar to the main residual stream, the model can process "speculative" residual streams which can be optimized to approximate the residual streams of future tokens, extending beyond immediate next-token prediction. By conditioning immediate next token prediction on speculative streams as well as previous context, the model gains the ability to predict upcoming tokens with a richer contextual scope. As a result, this approach mitigates the risks of overly greedy decoding, providing a more informed and contextually aware generative process.

Our goal is to develop an end-to-end trainable, single-model framework that integrates future token planning, enhances generation quality, and scales efficiently across multiple downstream applications. We propose following modifications to achieve these objectives. (a) Speculative stream design and initialization as described in Section 3.1.1 (b) Parallel speculation and verification as described in Section 3.1.2 (c) Parallel tree draft pruning, described in Section 3.1.3 and finally (d) Training objective as described in Section 3.1.4.

### 3.1.1 STREAMS DESIGN AND INITIALIZATION

Parameter efficient supervised fine-tuning (Hu et al., 2022) of decoder-only pre-trained language models involves training low-rank adapters to predict next target token $y_t$ given context tokens $(x_1....x_m)$ and previous target tokens $(y_1..y_{<t})$ on downstream applications. Although effective, this objective generates each token greedily and lacks a sense of future token planning (Qi et al., 2020) which may lead to sub-optimal generation quality (see Section 4.1.2). To inherently embed a notion of future token planning, we modify the training objective of the target model from next token prediction to n-gram prediction using multi-stream attention. This objective facilitates proactive token planning and mitigates over-fitting to local correlations (Yang et al., 2019; Qi et al., 2020). Furthermore, we extend this framework by sharing the key/value cache across all streams, allowing each of the $\gamma$ streams to generate speculative tokens with negligible latency overhead when the model is memory-bound. Specifically, each added stream predicts $p(y_{t+j}|y_{<t}, x)$, where $1 <= j <= \gamma$, while main stream predicts $p(y_t|y_{<t}, x)$.

In lossless mode, attention mechanism of main stream remains same as the standard multi-head attention mechanism (Vaswani et al., 2017) while in shared mode, we enable the main stream to attend to speculative streams, allowing it to plan its residual transformations based on anticipated future residual states by modifying the attention mechanism as

$$M_t^{k+1} = \text{MHA}(M_t^k, M_{\leq t}^k \oplus S_{t1...\gamma}^k, M_{\leq t}^k \oplus S_{t1...\gamma}^k) \tag{1}$$

where $M_t^k$ and $S_t^k$ refer to main and speculative streams at time step $t$ and layer $k$ and $MHA(H, H, H)$ denotes attention between query $HW^Q$, key $HW^K$ and value $HW^V$ as described in (Vaswani et al., 2017). On the other hand, each speculative stream $j$ at time step $t$ attends to previous main stream hidden states and previous speculative stream hidden states as:

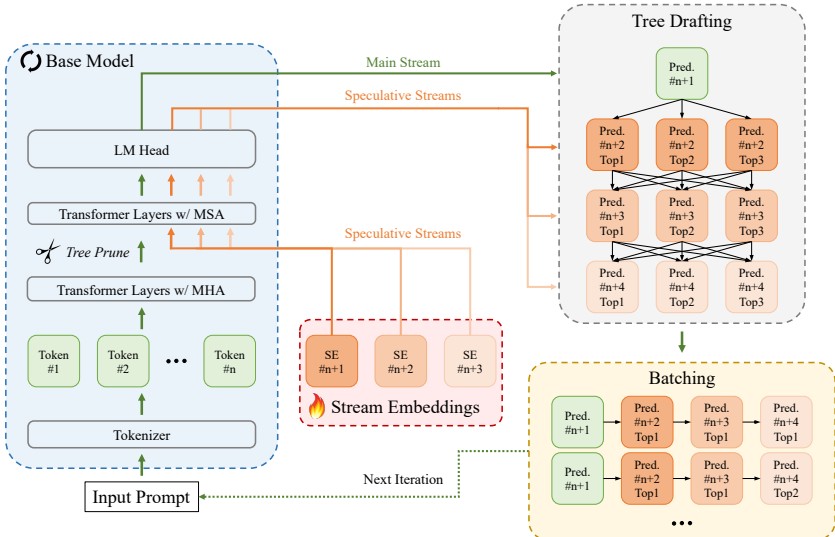

Figure 2: Architecture: We replace top $N_s$ multi-head attention (MHA) layers of the base model with multi-stream attention (MSA) layers as described in (2). Speculative streams are initialized using hidden states of layer $N - N_s$ and stream identifier embeddings (SE), as described in (3) and used to generate speculative draft in the form of a tree. The speculative tree draft from the previous iteration is batched for verification and pruned before stream insertion. During each forward pass previous tree draft is verified and a new tree draft is issued using speculative streams as described in 3.1.2

$$S_{tj}^{k+1} = \text{MHA}(S_{tj}^k, M_{\leq t}^k \oplus S_{t(\leq j)}^k, M_{\leq t}^k \oplus S_{t(\leq j)}^k) \tag{2}$$

Hidden state of last transformer layer $N$, $M_t^N$ is used to predict $y_t$, whereas each speculative stream at last layer, $S_{tj}^N$ predicts $y_{t+j}$. We refer to layers incorporating the attention mechanism in (Vaswani et al., 2017) as MHA layers while layers incorporating Equation (1) and Equation (2) are referred to as MSA layers. It is worth noting that the attention mechanism of speculative streams remains consistent across both shared and lossless modes. Key/value projections of main stream hidden states are cached during inference to avoid re-computation, whereas, we design speculative stream attention to specifically avoid storing additional key/value projections associated with individual streams. This is because speculative streams are trained to learn contextual features from main stream key/value context allowing us to not introduce additional caching overhead and operate within memory bounds of resource-constrained devices during inference. We initialize hidden states of speculative streams at layer $N - N_s$ instead of initializing them from the embedding layer, where $N_s < N$. Specifically, stream $j$ at time $t$ is initialized at layer $N - N_s$ as,

$$S_{tj}^{N-N_s} = f_\eta(M_t^{N-Ns}) + P_j^{N-N_s} \tag{3}$$

where $P_j$ is a stream identifier embedding that embeds a sense of relative position into streams and distinguishes the computation from main stream. $f_\eta$ is a linear transformation of rank $\eta$ to transform main stream hidden states into speculative stream hidden states. This initialization helps to reduce computation per forward pass, since only the main stream needs to be passed through $N - N_s$ layers, while speculative streams are passed through the last $N_s$ layers, decreasing the speculative FLOPs contribution by $(N - N_s)/N$ and in turn helping with peak power consumption on the device. In terms of forward pass latency, FLOPs do not contribute significantly when the model is memory bound, however, as we describe in Section 3.1.2, we sample additional tokens to make the model compute-bound, therefore FLOP reduction becomes crucial. We also experimented with value rotation based stream design which does not require identifier embeddings and incurs no parameter overhead as described in Appendix C.3.

### 3.1.2 PARALLEL SPECULATION AND VERIFICATION

In standard draft-target speculative decoding (Leviathan et al., 2023), speculation and verification processes happen sequentially. Speculative Streaming makes this process efficient by parallelizing speculation and verification. In each forward pass, the draft generated in the previous step is verified and a new draft is generated as shown in Figure 2. For instance, in step $s$, if draft tokens $(\tilde{y}_1..\tilde{y}_\delta)$ are accepted where $0 < \delta \le \gamma$, main stream $M_\delta$ is used to issue a correction token and logits from speculative streams $S_{\delta(1...\gamma)}$ are used to generate draft for step $s+1$.

Instead of using a linear sequence of speculated tokens for verification, we sample a tree of tokens from main and speculative streams, such that each path in the tree is one possible verification candidate. Tree drafting enables accepting the longest matching candidate sequence and more tokens can be advanced during each forward pass. To create a tree draft, instead of sampling 1 token from logits of speculative streams, $(z_1...z_\gamma)$, we sample top $k$ tokens and form a tree of sampled tokens as shown in Figure 2, such that tokens sampled from stream $n$ are predecessors of tokens sampled from stream $n+1$. We process a tree draft of speculative tokens in one forward pass by creating an additive attention mask (Vaswani et al., 2017) such that each node in the tree attends to its predecessor. Attention mask between $k^{th}$ token sampled from logits of stream $j$, $\tilde{y}_{jk}$ and the $m^{th}$ token sampled from logits of stream $n$, $\tilde{y}_{nm}$ is

$$a_{\tilde{y}_{jk}\tilde{y}_{nm}} = \begin{cases} 0 & \text{if j = n+1,} \\ -\infty & \text{otherwise} \end{cases} \qquad (4)$$

Please refer to Figure 12 for more details.

### 3.1.3 PARALLEL TREE PRUNING

One of the issues with the naive creation of a speculative tree draft is that every permutation between $k$ tokens sampled from each stream needs to be considered as a viable speculative candidate for the next verification pass. For instance, sampling $k$ tokens from each of $\gamma$ streams results in tree draft of size $1 + \sum_{g=1}^{\gamma} k^g$. Furthermore, each of the draft tokens is batched with $\gamma$ speculative streams in MSA layers to ensure that the generation of the next draft happens in the same forward pass, resulting in a batch size of $(1 + \gamma) * (1 + \sum_{g=1}^{\gamma} k^g)$. As batch size increases, target model inference becomes compute-bound, obviating the latency benefit of sampling more tokens. We mitigate this problem by introducing a parallel tree draft pruning layer, which prunes less probable tokens from the input tree draft based on transition probability between parent and immediate child tokens. To obtain transition probabilities without using proxy models, we use an early-exiting-based technique. Specifically, hidden states of the main stream at layer $l$, $M^l$ are passed through a low-rank linear transformation $o_\theta$, where the rank $\theta$ is typically set to a small value like 8 to keep parameter overhead minimal. We use original language modeling head, $H$ to obtain early exit logits, $\tilde{z} = H(o_\theta(M^l))$. $\tilde{z}_{pc}$ is used to approximate transition probability between parent token $p$ and child token $c$. The pruning layer can be inserted at any point in the network, guided by the trade-off between forward pass latency and pruning accuracy. Early insertion reduces latency but risks pruning potentially valuable tokens. Conversely, late insertion retains more "good" tokens but comes at the cost of increased forward pass latency. In all experiments described in Section 4.1, we insert the pruning layer just before speculative stream insertion. More details can be found in Appendix Figure 11.

### 3.1.4 TRAINING

Our supervised fine-tuning procedure entails training the base model on both the prediction loss of the next token and $\gamma$ future tokens. The overall loss function is defined as follows:

$$L_{ss} = -\alpha_0 (\sum_{t=1}^{T} \log p_\theta(y_t|y_{<t}, x)) - \sum_{j=1}^{\gamma} \alpha_j (\sum_{t=1}^{T-j} \log p_\theta(y_{t+j}|y_{<t}, x)) \qquad (5)$$

where $\alpha_0$ and $\alpha_j$ are set empirically to normalize losses of the next token and speculative tokens prediction using LoRA (Hu et al., 2022). Although training with Speculative Streaming is relatively cheap (see Appendix E), naive training increases batch dimension along sequence length axis by $\gamma$ causing attention computation to hit peak memory with larger batches. We employ a segment based attention method that helps reduce peak memory consumption and increases training throughput

significantly by dividing training sample into prompt and multiple completion segments. More details on segment attention can be found in Appendix D. Finally, Tree-pruning adapter described in Section 3.1.3 is trained on the next token prediction loss.

### 3.1.5 SHARED VS. LOSSLESS MODES

We investigate two primary deployment scenarios for mainstream LLMs. In generic chatbot-like use cases, pre-trained LLMs are instruction-tuned, and it is crucial to maintain the base model's output distribution while achieving speedup. For such cases, our approach operates in a lossless mode, where no trainable parameter sharing occurs between the main and speculative stream residual transformations ($\alpha_0 = 0$ in Equation (5)), and the main stream's attention mechanism remains unchanged, as outlined in Section 3.1.1. Conversely, in application-specific scenarios, adapter parameters are shared between the main and speculative streams, and the main stream's attention mechanism is modified (Equation (1)) to enhance task-specific response quality.

### 3.1.6 ACCEPTANCE CRITERIA

We adopt the rejection sampling-based acceptance criterion proposed by (Chen et al., 2023) to mitigate distributional shift between the draft and target models. Specifically, we apply rejection sampling to select tokens from each path in the pruned tree (see Section 3.1.3), and the longest accepted path is used to advance decoding. To adhere to the principles of rejection sampling, we replace the draft model's output distribution by introducing a virtual distribution, which leverages "prophet streams". More concretely, we replace the draft distribution $p(x \mid x_1, \ldots, x_{n+t-1})$ in Algorithm 2 of (Chen et al., 2023) with an augmented distribution $q(x \mid x_1, \ldots, x_n, s_{n0}, \ldots, s_{n(t-1)})$, where $s$ represents the state from the prophet streams. Thus, our acceptance criterion is formulated as follows:

$$r < \min\left(1, \frac{q(x \mid x_1, \ldots, x_{n+t-1})}{q(x \mid x_1, \ldots, x_n, s_{n0}, \ldots, s_{n(t-1)})}\right),\tag{6}$$

where $p$ and $q$ represent the draft and target distributions from (Chen et al., 2023), $r \sim U[0, 1]$, and $1 \leq t \leq \gamma$.

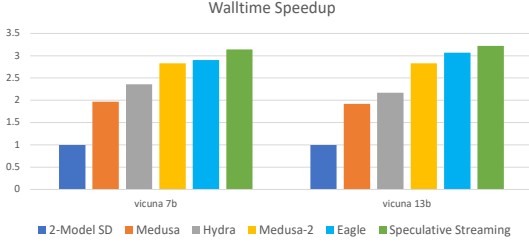
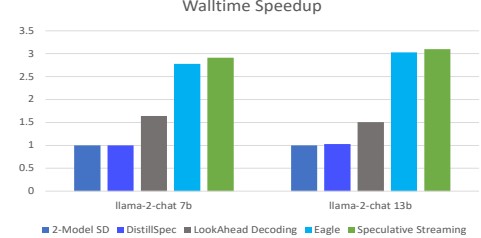

Figure 3: Mean walltime speedup on Vicuna models of various sizes to demonstrate scalability and generalizability of our approach on MT-Bench.

Figure 4: Mean walltime speedup on Llama-2 models of various sizes to demonstrate scalability and generalizability of our approach.

## 4 EXPERIMENTS

We evaluate our methods on a diverse set of downstream applications as well as generic reasoning oriented conversational tasks using pre-trained models of various scales.

**Datasets.** To test our method on user-facing application specific settings that are vital to on-device AI assistants we use a diverse set of tasks namely Text Summarization, Structured Queries and Meaning Representation using Dialogsum (Chen et al., 2021) dataset, the sql-create-context dataset built from WikiSQL (Zhong et al., 2017) and SPIDER (Yu et al., 2018), and e2e-nlg dataset (Dušek et al., 2020) respectively. Along with application specific settings, to test generalizability of our method, we evaluate on reasoning oriented chat-bot like setup using the multi-turn dialogue dataset, MT-bench (Zheng et al., 2023).

Table 1: Comparison of walltime speedup, tokens/step, and parameter overhead across models of different scales fine-tuned for downstream tasks. tokens/step indicates accelerator agnostic speedup metric. Metrics include exact match accuracy for SqlContext and Rouge for Dialogsum and E2E-NLG. Medusa and Speculative streaming parameters are fine-tuned jointly with the base model, while the base model is frozen during Eagle fine-tuning to to prevent adverse effects on generation metrics.

| Dataset | Model | Method | SpeedUp (↑) | Tokens/step (↑) | Metric (↑) | # Extra Parameters (↓) |
|---|---|---|---|---|---|---|
| SqlContext | Mistral-Instruct-7B | Baseline | 1.00 | 1.00 | 84.16 | − |
| | | Medusa-2 | 2.79 | 3.18 | 84.18 | $5.9E8$ |
| | | Eagle | 2.75 | 3.58 | 84.16 | $2.4E8$ |
| | | SS (ours) | **2.93** | **3.67** | **84.50** | $\underline{8.2E4}$ |
| | PHI-3-Instruct-3.8B | Baseline | 1.00 | 1.00 | 80.92 | − |
| | | Medusa-2 | 2.54 | 2.81 | 81.07 | $4.3E8$ |
| | | Eagle | 2.62 | 3.37 | 80.92 | $1.3E8$ |
| | | SS (ours) | **2.92** | **3.65** | **84.10** | $\underline{6.1E4}$ |
| | Llama2-7b | Baseline | 1.00 | 1.00 | 85.37 | − |
| | | Medusa-2 | 2.52 | 2.98 | 85.31 | $5.9E8$ |
| | | Eagle | 2.59 | 3.31 | 85.37 | $2.4E8$ |
| | | SS (ours) | **2.81** | **3.57** | **85.93** | $\underline{8.2E4}$ |
| DialogSum | Mistral-Instruct-7B | Baseline | 1.00 | 1.00 | 44.74/36.76 | − |
| | | Medusa-2 | 1.89 | 2.05 | 44.78/36.95 | $5.9E8$ |
| | | Eagle | 1.95 | 2.56 | 44.74/36.76 | $2.4E8$ |
| | | SS (ours) | **2.04** | **2.96** | **44.89/37.09** | $\underline{8.2E4}$ |
| | PHI-3-Instruct-3.8B | Baseline | 1.00 | 1.00 | 46.08/38.28 | − |
| | | Medusa-2 | 2.15 | 2.26 | 45.82/37.78 | $4.3E8$ |
| | | Eagle | 2.05 | 2.31 | 46.08/38.28 | $1.3E8$ |
| | | SS (ours) | **2.32** | **2.85** | **46.30/38.32** | $\underline{6.1E4}$ |
| | Llama2-7b | Baseline | 1.00 | 1.00 | 44.90/37.0 | − |
| | | Medusa-2 | 1.76 | 1.95 | 44.17/37.02 | $5.9E8$ |
| | | Eagle | 1.86 | 2.57 | 44.90/37.0 | $2.4E8$ |
| | | SS (ours) | **1.90** | **3.05** | **45.0/37.85** | $\underline{8.2E4}$ |
| E2E-NLG | Mistral-Instruct-7B | Baseline | 1.00 | 1.00 | 67.82/48.99 | − |
| | | Medusa-2 | 2.78 | 3.19 | 67.74/48.85 | $5.9E8$ |
| | | Eagle | 2.85 | 3.52 | 67.82/48.99 | $2.4E8$ |
| | | SS (ours) | **2.93** | **3.67** | **68.37/49.09** | $\underline{8.2E4}$ |
| | PHI-3-Instruct-3.8B | Baseline | 1.00 | 1.00 | 68.72/49.31 | − |
| | | Medusa-2 | 2.39 | 2.63 | 68.41/49.08 | $4.3E8$ |
| | | Eagle | **2.42** | **2.76** | 68.72/49.31 | $1.3E8$ |
| | | SS (ours) | 2.36 | 2.72 | **69.38/50.22** | $\underline{6.1E4}$ |
| | Llama2-7b | Baseline | 1.00 | 1.00 | 69.47/49.54 | − |
| | | Medusa-2 | 2.82 | 3.19 | 69.41/49.44 | $5.9E8$ |
| | | Eagle | 2.79 | 3.26 | 69.47/49.54 | $2.4E8$ |
| | | SS (ours) | **2.89** | **3.38** | **69.52/49.93** | $\underline{8.2E4}$ |

Table 2: Walltime latency (per sample) and auto-regressive calls comparison with standard draft-target (Two-model) speculative decoding approach using OPT-125m as the draft model.

| Dataset | Target | Method | Target calls | Draft Calls | Walltime Latency ($ms$, ↓) | Metric (↑) |
|---|---|---|---|---|---|---|
| SqlContext | OPT-1.3b | Two-model SD | 6.59 | 22.35 | 269.24 | 84.98 |
| | | SS (ours) | 7.79 | 0 | **133.48** | **87.40** |
| | OPT-6.7b | Two-model SD | 6.60 | 22.41 | 301.10 | 89.13 |
| | | SS (ours) | 6.88 | 0 | **157.04** | **89.34** |
| Dialogsum | OPT-1.3b | Two-model SD | 11.65 | 42.59 | 493.59 | 43.40/35.60 |
| | | SS (ours) | 13.41 | 0 | **248.26** | **44.07/35.99** |
| | OPT-6.7b | Two-model SD | 12.15 | 35.76 | 555.99 | 44.40/36.60 |
| | | SS (ours) | 14.45 | 0 | **444.67** | **44.42/36.81** |
| E2E-NLG | OPT-1.3b | Two-model SD | 8.86 | 31.47 | 345.72 | **69.48**/50.17 |
| | | SS (ours) | 9.80 | 0 | **164.23** | 69.32/**50.51** |
| | OPT-6.7b | Two-model SD | 8.90 | 31.58 | 412.02 | 69.34/**49.88** |
| | | SS (ours) | 10.31 | 0 | **244.80** | **69.45**/49.78 |

**Model Configuration.** We tested four different open source models of various scales, Phi-3-mini-4k-instruct(3.8B)(Abdin et al., 2024), Llama-2(7B)(Touvron et al., 2023b), Mistral(7B) (Jiang et al., 2023) and OPT(1.3B, 6.7B) (Zhang et al., 2022) on application specific settings. To test scalability of our approach we use Vicuna Models (7B, 13B) (Chiang et al., 2023) and Llama-2 chat models (7B, 13B). We compare our method with the draft-target speculative decoding methods (Leviathan

et al., 2023; Zhou et al., 2023) and single-model speculative decoding frameworks, Medusa (Cai et al., 2023), LookAhead decoding (Fu et al., 2023), Hydra (Ankner et al., 2024) and Eagle (Li et al., 2024). For the standard draft-target approach, we use OPT-125m, the smallest configuration of available open-source OPT models as the draft model.

**Metrics.** In application specific settings, we report wall-time speedups and generation quality metrics on held-out test set. We use Exact Match (EM) accuracy metric for the structured query task and Rouge1/RougeLSum metrics for the Dialog Summarization and Meaning Representation tasks. For generic chat-bot like settings, we train speculative stream adapters while keeping base model frozen as noted in Section 3.1.5 and report speedup and inference overhead.

**Inference.** Inference is performed using a batch size of 1 on a single Nvidia A100-80G GPU in float16 using greedy sampling and $T = 0$. Please refer to Appendix G for batching impact, Appendix C.4 for ablations on top-k sampling, $T = 1$ and Appendix H.1 for more experimental details. We set $N_s = 4$, $\gamma = 3$ and $k = 3$ for all experiments. Please refer to Appendix for hyperparameter ablations.

## 4.1 RESULTS

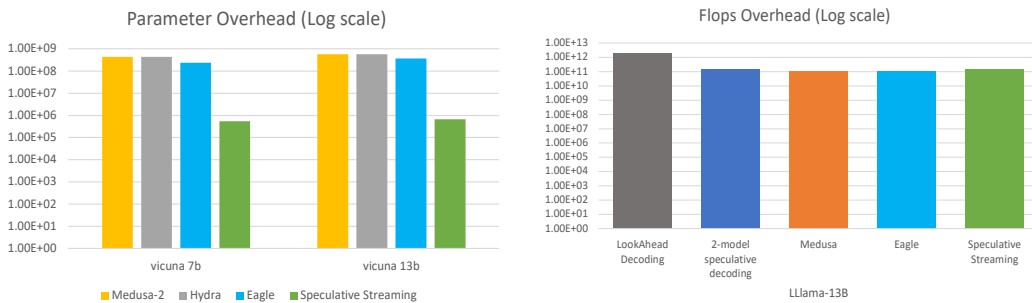

Figure 5: Parameter/Memory access overhead of different SD architectures with Vicuna models.

Figure 6: FLOP overhead of different SD architectures with Llama-13B.

### 4.1.1 EFFECTIVENESS

Table 1 compares the performance of our method against several baselines, including standard auto-regressive decoding, Medusa, and Eagle, in terms of speedup, tokens generated per step, and additional parameter overhead. Across a wide range of downstream tasks, Speculative Streaming consistently demonstrates superior wall-time speedups and tokens per step while incurring significantly lower parameter overhead compared to alternative approaches. As outlined in Table 2, our method also achieves lower wall-time latencies than conventional draft-target speculative decoding. This improvement arises because the marginal difference in target calls between the two approaches is insufficient to counterbalance the overhead introduced by auto-regressive drafting. For a deeper analysis, please refer to Appendix H. Furthermore, it is important to highlight that the generation quality of Speculative Streaming consistently outperforms that of next-token prediction-based fine-tuning, positioning it as a compelling alternative to LoRA-based fine-tuning approaches. The speedup gains of our approach remain consistent across multi-turn conversational tasks evaluated on MT-Bench. In lossless settings, our method consistently achieves better speedup than alternative approaches across Vicuna and Llama models of various scales (see Figure 3 and Figure 4), while incurring significantly lower memory access and computational overhead (see Figure 5 and Figure 6), demonstrating the generalizability and scalability of our approach.

### 4.1.2 WHY DOES IT WORK?

**Generation Metrics:** To investigate the improvements in generation quality achieved by our approach, we designed an experiment where the model predicts the next token while attending to a set of future $\gamma$ ground truth tokens beyond the next token. Our hypothesis was that by granting the model access to these future tokens, the attention mechanism would enhance its ability to anticipate and plan for the next token, thus improving generation quality. Specifically, we postulated that:

$$p(y_t = g_t | y_{<t}, y_{t+1..t+\gamma}, x) > p(y_t = g_t | y_{<t}, x) \tag{7}$$

Here, $g_t$ represents the ideal ground truth token that maximizes the generation quality metrics. To validate this hypothesis, we modified the attention mask, allowing the model's residual states to "peek" into future residuals. As shown in Figure 8, this modification led to significant improvements in generation metrics.

While such access to future tokens is not feasible during inference, where future states are unavailable, our approach enables the model to approximate future residual states using speculative streams. As demonstrated in Figure 7, these speculative streams, $S_{tj}$, progressively align with the true residual states of the next tokens as they propagate through the model layers. Crucially, our method allows the primary stream, $M_t$, to attend not only to the current context up to token $y_t$ but also to the speculative streams $S_{tj}$. This multi-stream attention mechanism refines the transformations within $M_t$, aligning them more closely with the context of the upcoming $\gamma$ tokens. As a result, the model effectively "plans" for future tokens, leading to measurable improvements in generation quality.

**Speedup:** Medusa attempts to generate the hidden states of speculative tokens $y_{(t+1...t+\gamma)}$ by applying a simple context independent transformation to the last hidden state of the current token $y_t$. However, this method has significant limitations. The absence of attention mechanisms results in lower similarity metrics between the speculative hidden states generated by Medusa and the true hidden states, which are obtained by feeding the actual next token into the model (see Figure 7). In contrast, our proposed technique leverages multi-stream attention, wherein speculative streams are allowed to attend to each other as well as to the main stream. As these streams propagate through the model layers, they more closely approximate the true hidden states of the actual next tokens, resulting in higher similarity, thereby increasing the acceptance rate of the speculated tokens.

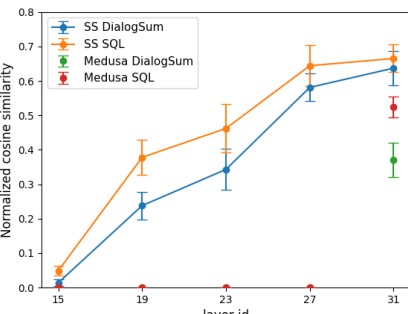

Figure 7: Cosine similarity between speculative residual states and residual state of ground truth tokens with Speculative Streaming and Medusa. As the streams propagate through the model, their representations become increasingly aligned with the ground-truth tokens in contrast to Medusa.

Figure 8: Generation performance of the Phi-3 model when trained to attend to $\gamma$ ground truth tokens beyond the immediate next token during prediction. Incorporating future ground truth tokens into the attention mechanism leads to substantial improvements in generation performance.

## 5 CONCLUSION

In this paper, we proposed Speculative Streaming, a method to accelerate decoding of large language models. Compared to the standard speculative decoding approaches, Speculative Streaming removes the need for an auxiliary "draft" model. Instead, it unifies speculation and verification by efficiently fusing multiple speculative streams into a single "target" model. Speculative Streaming simplifies the fine-tuning process and achieves better generation quality and speedup compared to previous approaches. It is also parameter efficient and removes the need for loading two models into the memory, making it a suitable approach for resource-constrained scenarios.

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

## A    ABLATIONS

We conducted extensive ablation studies to identify the optimal draft size and to evaluate the impact of tree pruning, as illustrated in Figure 9. Tree pruning enhances speedup by eliminating less probable speculative paths, thereby preventing the model from entering a compute-bound phase. Further details are provided in Appendix C. Additional ablations were performed to determine the ideal number of Multi-stream Attention (MSA) layers and their influence on fine-tuning performance, as well as the effects of value projection rotation and Top-k sampling. An increase in the number of MSA layers consistently improves generation metrics across all downstream tasks, supporting the hypothesis that Multi-Stream Attention facilitates effective planning. Our method also demonstrates robustness to non-greedy Top-k sampling, which is critical for maintaining diversity and quality control in generated text. Please refer to Appendix C for comprehensive results.

## B    IMPLEMENTATION DETAILS

### B.1    TREE DRAFT MANAGEMENT

In this section, we go into more detail of tree draft sampling, flattening, and pruning. As shown in the main paper, when processing prompt $(x_1...x_t)$, we insert speculative streams along with the last token to generate logits, $z_t$ corresponding to main stream and $(z_{t1}...z_{t\gamma})$ corresponding to speculative streams. Tree draft is sampled following the procedure described in Section 3.1.2. The sampled draft is then flattened along the sequence length dimension and the attention mask is composed such that child nodes attend to their predecessors starting with root as shown in Figure 11 and Figure 12. The root token of the tree draft is the correction issued by main stream. Each iteration after prompt processing involves verifying the previous tree draft and sampling a new one. After passing the tree draft through $N - N_s$ layers, we use contextual features learned by middle layers to approximate transition probability between parent and child tokens. As shown in Figure 11, since the transition probability between token "$parameter$" and "$compare$" is less than a set threshold, we prune the sub-tree starting from "$compare$" in the feature domain , and $m_2, m_5, m_6$ are pruned. Please note that the key value cache of layers $0..(N - N_s - 1)$ before the pruning layer is not trimmed at this point to keep pruning latency overhead minimal. Key value cache backtracking is done lazily after each generation step. Speculative streams are inserted alongside each node in the pruned draft. Layers $(N - N_s..N)$ use Multi-stream attention as described in Equation (1) and Equation (2). The verification procedure finds the longest matching path in the pruned tree that main stream can accept. As shown in Figure 11, path ("$parameter$", "$efficient$", "$speculative$") is accepted. Correction token sampled from logits of main stream corresponding to last accepted token, $m_1$ becomes new root while tokens sampled from logits of streams $(s_{10}, s_{11})$ form the sub-tree.

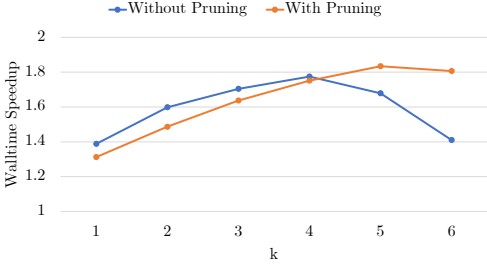

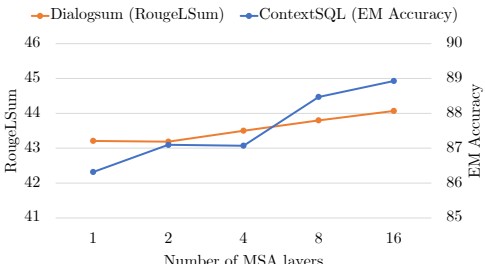

Figure 9: As more tokens ($k$) are sampled for tree drafting, speedup initially increases. This trend reverses as $k$ continues to increase as the model transits to the compute-bound phase. Pruning less probable paths helps reduce compute, offering more speedup.

Figure 10: As the number of multi-stream attention layers increases, metrics on downstream tasks improves. Typically $N_s = 2$ to $8$ yields a good trade-off between generation metrics and FLOPs overhead and training time.

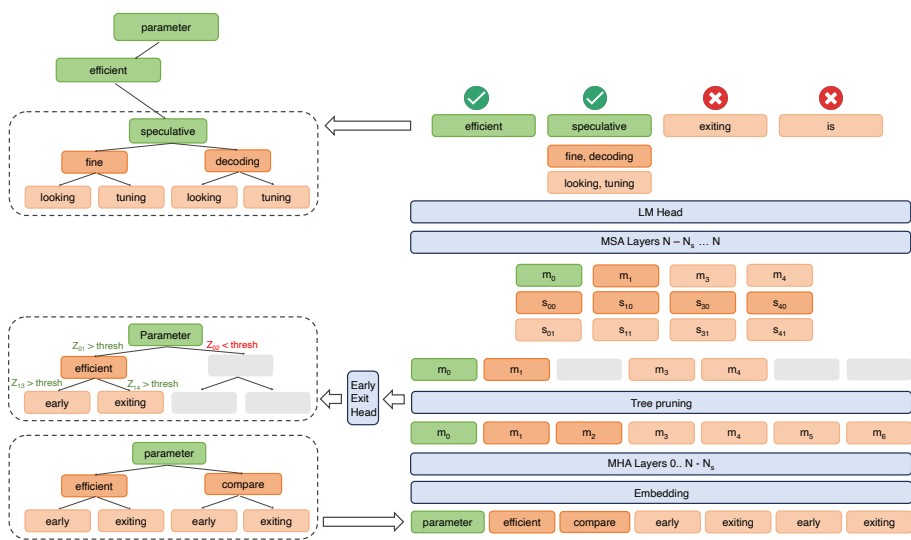

Figure 11: Parallel tree draft speculation and verification: Tree draft from the previous iteration is flattened for verification. After $N - N_s$ MHA layers, the tree pruning procedure obviates less probable tokens based on transition probability between parent and child tokens. In this illustration $Z_i$ denotes normalized early exit logits corresponding to main stream at index $i$, $m_i$, while $Z_{ij}$ denotes transition probability between token at index $i$ and $j$ in flattened tree draft. The verification procedure is subsequently run on the pruned tree and speculative tokens are sampled from streams corresponding to the latest accepted token. In above illustration, "$speculative$", "$fine, decoding$" and "$looking, tuning$" are sampled from streams $m_1$, $s_{10}$ and $s_{11}$.

## C    ABLATION:

### C.1    SPECULATIVE DRAFT SIZE.

To improve the acceptance rate of the tree draft, we try various settings of $\gamma$, the number of speculative positions, and $k$, the number of sampled tokens per speculative position. Figure 9 shows walltime speedup for $\gamma = 3$. As we sample more tokens from each speculative position, advancement per forward pass, $\beta$ increases since more candidates are available for verification, leading to more speedup. However, as we continue to increase $k$, forward pass latency overhead becomes more prevalent as the model transitions into compute-bound phase and the speedup reverses the course. This is because naively forming a tree draft leads to an exponential increase in batch size with $k$ as described in 3.1.3. We insert a tree pruning layer to remove less probable paths and reduce the size of the tree draft. Pruning tree draft reduces forward pass latency, and a well calibrated threshold ensures that only noisy paths in the tree get pruned. Tree pruning tends to help with walltime speedup as $k$ continues to increase as shown in Figure 9.

### C.2    NUMBER OF MSA LAYERS

There are trade-offs involved in deciding the number of MSA layers to incorporate in terms of downstream generation metric, training time, and FLOPs increase. As we increase the number of MSA layers, the generation metric improves and this trend remains the same across different downstream tasks. Typically incorporating MSA in the top 2 - 8 layers offers a good trade-off between metric, FLOPs increase and training time. Figure 10 shows the generation performance of the OPT-1.3b model on Structured Query and Summarization tasks.

Figure 12: Attention mask for tree draft is composed in such a way that child tokens can attend to all predecessors starting from root, root being correction issued by main stream. In this illustration, *"early"* attends to *"parameter"* and *"efficient"* and itself since *"parameter − efficient − early"* forms one path in tree. *"early"* is also replicated to form another path *"parameter − compare − early"*. This attention mask allows batching multiple paths and increasing acceptance rate as number of candidates increase.

## C.3 VALUE ROTATION

We analyzed more ways of differing computation of main stream from speculative streams. Apart from using dedicated stream embeddings, one way to differentiate the computation while incorporating a sense of relative position is simply rotating streams relative to each other. In this ablation, we initialize each stream with the main stream hidden state and rotate the value projection during attention computation in the proportion of the relative distance from main stream as :

$$V_{tn}^k = V_t^k e^{i\epsilon n} \tag{8}$$

Where $1 <= n <= \gamma$ is stream index, $V_t^k$ denotes value projection of main stream at time step $t$ and layer $k$, while $V_{tn}^k$ denotes value projection of stream n, $0 \leq \epsilon \leq \frac{\pi}{2N}$ denotes an arbitrary rotation step and $N$ denotes the sum of maximum sequence length and number of streams. Figure 13 (a) shows the effect of using value rotation on Rouge scores on the Dialog Summarization task with the Phi-1.3b model. Downstream metric for value rotation-based approach tends to be lower than using dedicated stream embeddings across different settings of MSA layers, however, the trend of increasing metric with added MSA layers remains the same. It is worth noting that for $N_s = 16$, simply rotating value projections achieve better metrics than using $N_s = 4$ with dedicated stream embeddings.

## C.4 TOP-K SAMPLING

In the main paper, we reported speedup results using greedy sampling and T=0. To further analyze speedups in the Top-k sampling regime, we try various values of $k$ and T = 1 for both Medusa style and Speculative Streaming approaches. Figure 13 (b) shows the effect of increasing $k$ on the walltime speedups and call reduction ratios[1]. Although increasing $k$ leads to lower wall-time speedups for both baseline and target methods due to stochastic rejection of tokens, our approach retains its lead achieving better call reduction ratios and walltime speedups across different values of $k$.

---

[1]The call reduction ratio represents the ratio of the number of 'model.forward()' calls required for autoregressive decoding to those required for speculative streaming. It is equivalent to the average number of tokens generated per 'model.forward()' call during target speculative streaming.

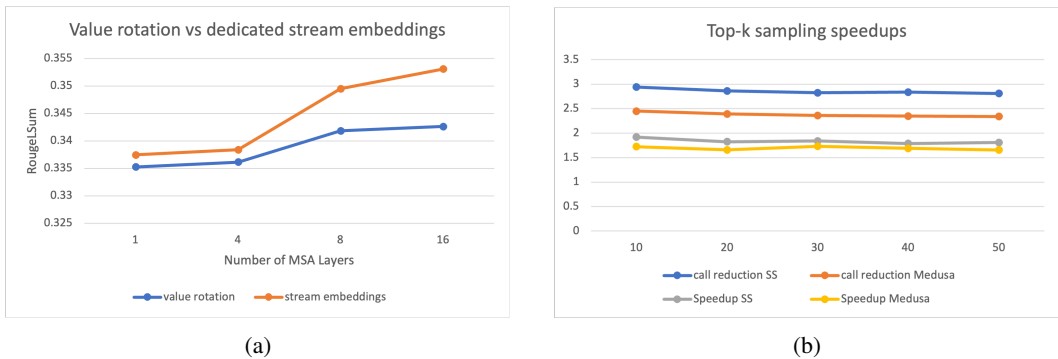

(a)                                                (b)

Figure 13: (a) We analyze the effect of value projection rotation on RougeLSum scores of the Dialog summarization task using PHI-1.3b as the base model for different numbers of MSA layers. Each stream is rotated in proportion to the distance from the main stream. (b) We study the effect of top-k sampling on wall-time speedups and call reduction ratios (mean tokens genearted per step) for Speculative Streaming (SS) and Medusa-style approaches using OPT-1.3b as a base model on the Meaning Representation task.

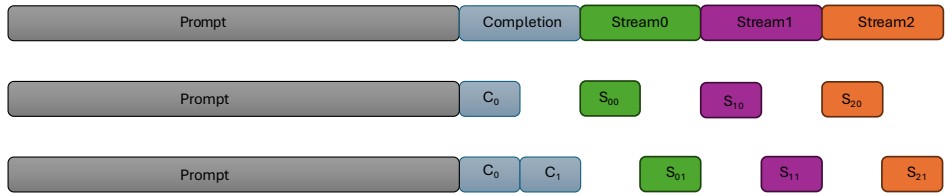

Figure 14: Streams corresponding to prompt are not required while training. Completion is divided into multiple segments and streams of each segment only attend to previous streams from same segment and main stream of previous segments. Uncolored portion indicates those tokens/streams are not required to be kept in memory.

## D  SEGMENT ATTENTION

Naive training with speculative streaming increases the batch dimension along the sequence length axis by a factor of $\gamma$, resulting in attention computation reaching peak memory usage with larger batches. To address this issue, we propose a segment-based attention method that significantly reduces peak memory consumption while enhancing training throughput. We divide each training sample into a prompt and multiple segments of completion. Since each stream corresponding to each token must attend to the previous streams of the same token as well as to the prompt tokens, we can eliminate the need for prompt streams in our design. Furthermore, by segmenting the completion, we retain only the streams associated with the required segments in memory, as illustrated in Figure 14. This design significantly reduces peak memory consumption and ensures the scalability of our approach when training with larger batch sizes, ultimately yielding improved throughput.

## E  TRAINING COST

Since speculative streaming is parameter efficient, training involves fine-tuning only LoRA parameters of MSA layers and it's comparable to training Medusa heads. We finetuned Vicuna-7B model on the ShareGPT dataset in $\sim 5$ hours using segment attention, comparable to the 3-4 hours required for training Medusa heads. We also managed to train 33B Vicuna models on a single 80-GB GPU by loading the base model in nf-4 precision and keeping only the adapters of 4 MSA layers in full precision.

## F    Compute and Memory Profiling

The draft overhead associated with the standard draft-target speculative decoding approach tends to be non-trivial especially when the latency ratio between target and draft models $c_{target}/c_{draft} <= 10$. This is because speculation and verification procedures are run in serial manner. Figure 15 shows the kernel utilization timeline when OPT-125m is used as a draft while OPT-1.3b model is used as the target. Auto-regressive draft generation decreases overall kernel utilization in draft-target approach, while additional computation involved in MSA layers increase kernel utilization in case of Speculative Streaming (see Figure 17) thereby efficiently utilizing the accelerator and speeding up the decoding process. Negligible cost draft models may offer a better choice to keep kernel utilization at higher levels in case of draft-target approach, however, acceptance rates tend to drop as draft model size decreases.

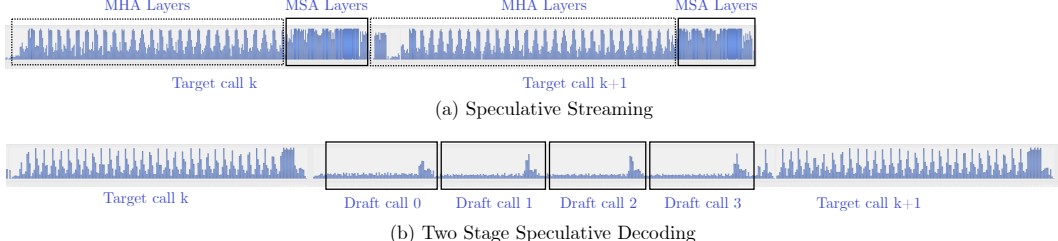

Figure 15: Kernel utilization timeline for speculative streaming and the standard draft-target speculative decoding. Draft-target approach runs speculation and verification in serial manner while it is parallelized in Speculative Streaming. Auto-regressive draft generation often has low kernel utilization as shown leading to decreased overall kernel utilization while MSA layers in Speculative Streaming increase kernel utilization by generating a non-autoregressive draft and speeding up decoding significantly.

## G    Batching

All the results presented in Section 4 are with batch size of 1 for on-device setup. We also experiment with batching for server setup where queries from multiple users are batched to increase throughput and accelerator utilization. To achieve maximum throughput with batching, we disable tree decoding and tree pruning and use only best speculated path for each decoding step for every sequence in a batch. Since our method primarily relies on utilizing flops to accelerate decoding, with batching we do see some degradation in speedup per sample as depicted in Figure 16, however we consistently achieve >2X speedups while keeping throughput same as batched autoregressive decoding.

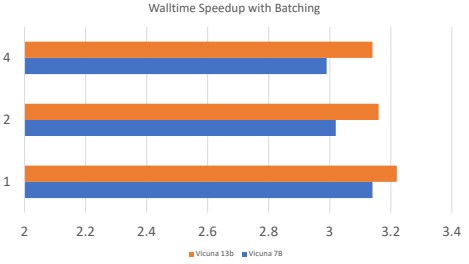

Figure 16: Walltime speedup for different batch sizes with Vicuna Models.

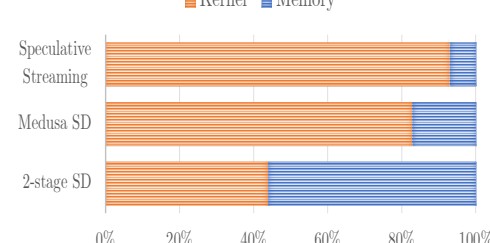

Figure 17: Kernel and Memory utilization comparison on Nvidia A-100.

## H    Analysis of 2-model speculative decoding

Speculative Streaming consistently achieves significantly lower walltime latency than standard draft-target speculative decoding as depicted in Table 2. It is worth noting that, target model calls of

draft-target speculative decoding are slightly lower than Speculative Streaming, however, it comes at the cost of auto-regressively running draft model $\gamma$ times to generate speculative draft. On the other hand, draft generation with Speculative Streaming incurs almost no additional latency overhead, as target model decoding tends to be memory-bound even with increased tree draft size. This translates to increased kernel utilization and arithmetic intensity as shown in Figure 17.

An argument could be made that a smaller draft model may perform better since drafting should cost less, but acceptance rates may drop as well as the draft model size is decreased. To formalize the comparison with standard draft-target speculative decoding, we do the following analysis, let's say, $C_{draft}$ is the latency cost associated with forward pass through the draft model, $C_{target}$ is the cost associated with forward pass through target model, while $C_{ss}$ is cost associated with speculative streaming forward pass. $\zeta$ is the number of decoding tokens advanced during the verification step for the draft-target approach while $\beta$ is the number of tokens advanced in Speculative Streaming. We equate latency cost associated with single token advancement to compare both approaches.

$$(\gamma * C_{draft} + C_{target})/\zeta = C_{ss}/\beta \qquad (9)$$
$$(\gamma + C_{target}/C_{draft})/\zeta = (C_{ss}/C_{draft})/\beta$$

Assuming $\gamma = 4, C_{target}/C_{draft} = 10$, and $C_{ss} \approx C_{target}$, $\zeta = 1.4\beta$, meaning that advancements per verification step in standard draft-target approach have to be 1.4X of Speculative Streaming to achieve wall time latency parity. Note that, this analysis ignores cache adjustment overhead and prompt processing overhead, but provides valuable intuition to guide the choice between draft-target vs Speculative Streaming approaches. We also analyze under which settings speculative streaming is likely to offer more benefits as compared to the standard draft-target approach. Fig. 1b shows theoretical speedups of Speculative Streaming over draft-target based approach for different Target to draft latency ratios. As the latency ratio increases, the draft-target approach is likely to offer more speedup benefits when $\zeta/\beta > 1$, meaning that when the draft model is accurate enough to achieve more token advancements per target model verification step than Speculative Streaming and also small enough to yield higher latency ratios, it is likely to benefit more. Finding/creating such a model usually requires significant engineering efforts. In downstream application settings, finding ideal draft models becomes even more challenging since $\zeta$ tends to vary based on application. If applications share the draft model and only train adapters, the draft model may not remain small enough to meet target-to-draft latency ratios, making it challenging to achieve more speedups than Speculative Streaming.

## H.1 EXPERIMENTAL SETUP DETAILS

For experiments described in 4, our recipe involves training LoRA adapters for 5 epochs on the downstream datasets in BFloat16, using the AdamQ optimizer, a learning rate of 5e-4, and a linear scheduler. For tree pruning (see Section 3.1.3), we use a low-rank linear transformation of rank 8 to keep parameter overhead minimal. We set $\alpha_0 = 1$ and $\alpha_j = 0.1$ for $j = 1...\gamma$ to weigh speculative loss relative to next token prediction loss. We experimented with linear transformations of different ranks to initialize speculative streams from main stream as described in Equation (3), however we find that simply using identity transformation achieves similar performance with much less parameter overhead. We use identity transformation for all the experiments described in Section 4. We report best results for Medusa and our approach over different $\gamma$ and $k$ values. For speculative tree draft generation, we used the optimal settings for both Medusa and Speculative Streaming, specifically $\gamma = 3$ and $k = 4$. We pass 32 nodes as a tree draft for speculative streaming after the pruning layer while in case of Medusa we pass 64 nodes, as these configurations yield the best wall-time speedups for respective approaches. We also report accelerator agnostic speedups (mean tokens generated per step) assuming negligible verification and draft composition overhead as latency of forward pass, verification and draft composition procedures vary greatly depending on accelerator (*e.g.* a mobile device neural engine *vs.* Nvidia A100), while tokens/step metric tends to serve as roof-line for achievable speedup. Lastly, we use "hard" matching criteria for verification of speculative draft. Relaxing this criteria to "soft" matching may yield higher speedups (Cai et al., 2023). To compare with Medusa (Cai et al., 2023) style approach, we use pre-trained base models with LoRA adapters (Hu et al., 2022) of rank 32 and Medusa heads as the baseline, and Speculative Streaming with the same base models, stream embeddings and LoRA adapters as target. Medusa heads are

trained following the recipe described in (Cai et al., 2023). Both Medusa heads and the number of maximum streams are fixed to 4 and the residual blocks per head used in Medusa are set to 1. For comparison with standard draft-target speculative decoding (Leviathan et al., 2023), we use OPT models since they come with different configurations and sizes. OPT-125m is deployed as a draft model while OPT-1.3b and OPT-6.7b are used as target models since a ratio of 10-100X is typically considered to be optimal. We compare our approach with LookAhead decoding using best configuration reported in (Fu et al., 2023).

## I    PARAMETER OVERHEAD

In terms of parameters, each Medusa head adds about $h^2 + hv$ parameters, where $h$ is the hidden size and $v$ is the vocabulary size. The number of Medusa heads also scales linearly w.r.t. $\gamma$, the length of the speculative window, which in turn increases parameter overhead linearly with $\gamma$. On the other hand, Speculative Streaming uses speculative adapters which do not scale with $\gamma$. Although, Stream identifier embeddings scale with $\gamma$, the parameter overhead associated with each embedding is linear to $h$. Furthermore, in fine-tuning settings "speculative adapter" parameters are shared with base model adapters, therefore, parameter overhead associated with our approach is just $\gamma h$.

## J    ADDITIONAL RELATED WORKS

The inference speed of large language models (LLMs) is often constrained by the sequential nature of auto-regressive decoding, which requires a complete forward pass of the network for each token generated. To mitigate the high inference latency, various strategies have been proposed to reduce the memory footprint of LLMs. Techniques such as model quantization (Frantar et al., 2022; Yao et al., 2022; Dettmers et al., 2023), knowledge distillation to smaller models (Gu et al., 2023; Agarwal et al., 2023), and pruning (Frantar & Alistarh, 2023; Sun et al., 2023a) have emerged as effective solutions. More recently, Confident Adaptive Language Modeling (CALM) (Schuster et al., 2022) has introduced a method to dynamically adjust computational resources per token through early exiting in decoder layers. While CALM shows promise, it is hindered by issues related to key-value (KV) mismatch (Corro et al., 2023). To address the KV mismatch problem, skip decoding (Corro et al., 2023) allows for the bypassing of an increasing number of layers based on the position in the decoded sequence. While this approach eliminates KV mismatch, the predefined restrictions on the number of layers bypassed can lead to suboptimal generation quality. In contrast, speculative decoding methods provide a significant advantage over dynamic computing approaches, as they maintain generation quality while enhancing inference efficiency.

## K    LONG CONTEXT EXPERIMENTS

To evaluate performance on long sequences, we trained speculative adapters on the Arxiv-summarization dataset (Cohan et al., 2018) and tested it on the Summarization task from the LongBench dataset (Bai et al., 2023). Since the KV cache is shared between the main and speculative streams, there is no additional runtime memory overhead. While compute in attention layers increases due to longer context, the compute in MLP layers remains the same, and decoding is still memory bandwidth bound. We achieved a 2.64X speedup on the Summarization test set using gamma = 3 and k = 4. We will include these LongBench experiments in the Appendix of the final revision.

| k | Speedup |
|---|---------|
| 1 | 2.21 |
| 2 | 2.35 |
| 3 | 2.52 |
| 4 | 2.64 |
| 5 | 2.58 |

Table 3: Speedups for long-context summarization tasks with varying top-$k$ tokens sampled during drafting.

SELECT in _ count y

SELECT in _ count y _ tu ition _ per

SELECT in _ count y _ tu ition _ per_ credit _ credit _

SELECT in _ count y _ tu ition _ per_ credit _ hour __ fall _ _

SELECT in _ count y _ tu ition _ per_ credit _ hour __ fall _ 2009 _ FROM table _

SELECT in _ count y _ tu ition _ per_ credit _ hour __ fall _ 2009 _ FROM table _ 22 30 88 81 _

SELECT in _ count y _ tu ition _ per_ credit _ hour __ fall _ 2009 _ FROM table _ 22 30 88 81 _ 2 WHERE college = "

SELECT in _ count y _ tu ition _ per_ credit _ hour __ fall _ 2009 _ FROM table _ 22 30 88 81 _ 2 WHERE college = " Mer Er " College <\s>

SELECT in _ count y _ tu ition _ per_ credit _ hour __ fall _ 2009 _ FROM table _ 22 30 88 81 _ 2 WHERE college = " Mer Cer " <\s>

Figure 18: Speculative streaming on SQL generation task for $\gamma = 4$ and $k = 1$, each pass verifies the previous draft and generates a maximum of 5 tokens. For instance in pass 4, "$credit$" and "_" (shown in red) are rejected and "$hour$", "_", "$fall$", "_", "_" are speculated.

# Person 2 # and

# Person 2 # thinks Lincoln is a character

# Person 2 # thinks Lincoln was a character and he

# Person 2 # thinks Lincoln was a man of character and he

# Person 2 # thinks Lincoln was a man of sound character and # person

# Person 2 # thinks Lincoln was a man of sound character and # person 1 # adm ires him

# Person 2 # thinks Lincoln was a man of sound character and # person 1 # adm ires him for his courage and and

# Person 2 # thinks Lincoln was a man of sound character and # person 1 # adm ires him for his courage and rights and humility . 

Figure 19: Speculative streaming on Dialog Summarization task for $\gamma = 4$ and $k = 1$, each pass verifies the previous draft and generates a maximum of 5 tokens. For instance, in pass 3, "$is$", "$a$", "$character$" are rejected and "$was$", "$a$", "$character$", "$and$", "$he$" are speculated.

## L  QUALITATIVE EXAMPLES

In this section, we present qualitative examples to illustrate the effectiveness of Speculative Streaming. By examining specific instances, we aim to highlight how this approach enhances the overall performance of the decoding process. An example of the SQL query generation task is shown in Figure 18, while a dialog summarization example is shown in Figure 19. Each row indicates the previous sequence of accepted draft tokens (in black) and the new sequence of generated tokens in green/red. We use $\gamma = 4$ and $k = 1$ to illustrate the decoding process. Green tokens in each row indicate tokens accepted in the next forward pass, while red tokens indicate tokens rejected in the next forward pass. Speculative Streaming appears to generate meaningful drafts with high acceptance rates by capturing dependencies between tokens quite effectively, despite generating them in a non-auto-regressive manner.

