# OpenReview forum: "Speculative Streaming: Fast LLM Inference without Auxiliary Models"
_ICLR.cc/2025/Conference — Submitted to ICLR 2025_

### Official Review · Reviewer_hmCo · 2024-10-30

**Soundness:** 3
**Presentation:** 2
**Contribution:** 2
**Rating:** 3
**Confidence:** 4

**Summary:**

This paper introduces Speculative Streaming, a method that integrates multiple soft embeddings before the $N-N_s$ multi-head attention layers of target LLMs and fine-tunes LLMs using LoRA, enabling the parallel prediction of these additional positions. Compared with prior methods like Medusa which modifies the MLP layers, Speculative Streaming focuses on modifications of LLM attention, providing a unique and investigative perspective on speculative decoding. The approach’s efficacy is validated through evaluations across various tasks and LLMs, achieving a 1.9 to 3x overall speedup.

**Strengths:**

1. The paper presents a novel investigation to speculative decoding by modifying the attention mechanism. Compared to prior methods like Medusa and Eagle, the authors claim that Speculative Streaming requires fewer parameters while achieving superior speedup performance.
2. This work provides an insightful explanation for Speculative Streaming’s effectiveness, which is illustrated in Section 4.1.2. The findings suggest that by enabling LLMs to attend to future tokens, the attention mechanism would enhance its ability to anticipate and plan for the nearest next token, thus improving generation quality. This result holds promise for future research directions.
3. The authors provide extensive experimental details and analysis in the Appendix, covering aspects like the tree drafting process, training costs, parameter overhead, and etc. These additions effectively address some of my initial concerns.

**Weaknesses:**

1. **The original LLM distribution is altered**. Unlike most speculative decoding such as [1], Medusa-1[2], and Eagle[3] that maintains the original output distribution of target LLMs, Speculative streaming (SS) modifies the target LLM's attention, enabling the main stream to attend to speculative streams and thus altering the original output distribution. This modification would largely impact the application of this paradigm since the generation quality may not be guaranteed. Although comparative performance is shown in Tables 1 and 2, SS performance is likely sensitive to fine-tuning procedures and evaluation domains.
2. **Lack of Experimental Detail**. In the abstract and results of Section 4.1.1, the authors claim that in **lossless** settings for generic chatbot applications that **do not necessitate supervised fine-tuning**, SS could achieve 2.9 - 3.2X speedup while maintaining the integrity of the base model’s output. However, I found no detailed description of the experimental setup for this configuration. Given that SS introduces stream embeddings and LoRA tuning, clarification on how SS accomplishes lossless speedup without fine-tuning is necessary. Detailed settings should be provided. Besides, could SS maintain the same output distribution as the original LLM?
3. **Some demonstrations are confusing**. In Section 3.1.4 (Training) and Appendix D, the authors mention using LoRA to train the model with the objective in Equation (5). However, LoRA-tuned parameters are not highlighted in Figure 2, which suggests that only stream embeddings are tuned. A clear discussion is needed on which parameters are subject to LoRA tuning—does LoRA tuning apply to all multi-stream attention (MSA) layers?
4. **Computation overhead of SS**. In Table 1, SS is shown to require fewer parameters than Eagle and Medusa. However, SS appears to incur higher computation overhead in drafting than Medusa, as the extra stream representations require forward passes through the last $N_s$ MSA layers, including both the MLP and attention layers. This adds an additional computational burden to the whole framework.
5. **Clarification of Multi-Stream Attention**. The term "multi-stream attention" may be somewhat misleading. SS resembles a combination of soft-prompting (via stream embeddings) and LoRA tuning, with the drafting process involving not only MSA layers but also the MLP layers in the last $N_s$ layers. Consideration of alternative wording might improve clarity.

[1] Fast Inference from Transformers via Speculative Decoding. Leviathan et.al. ICML 2023.

[2] Medusa: Simple LLM Inference Acceleration Framework with Multiple Decoding Heads. Can et.al. ICML 2024.

[3] EAGLE: Speculative Sampling Requires Rethinking Feature Uncertainty. Li et.al. ICML 2024.

**Questions:**

Most of my primary concerns are outlined in the weaknesses section above. Here are additional minor concerns:

- In line 107, the phrase "Moreover, using a dedicated layer leads to significant parameter overhead" is somewhat misleading. Compared to Medusa, which incorporates multiple MLP heads, Eagle requires only one autoregressive head, thereby **reducing parameter overhead**, as shown in Table 1.
- The authors investigate a tree drafting strategy in Section 3.1.1 and Appendix A.1. Compared to Medusa’s fixed tree structure of 64 candidate tokens, SS increases the number of candidates to a batch size of $(1+\gamma) \cdot \left(1+\sum_{g=1}^\gamma k^g\right)$ (where $\gamma=3$ and $k=3$), while limiting exploration breadth and depth to 3. Have the authors compared this tree implementation with that of Medusa?
- In line 321, the authors mention that "For generic chatbot-like settings, where fine-tuning is not required, we train speculative streams while keeping the base model frozen as noted in Section 3.1.4." The specific settings for this configuration should be clarified: is LoRA tuning still applied, or are only the stream embeddings tuned?
- Please provide a more detailed explanation of the CR (call reduction) ratio metric. Does it correspond to the mean number of generated tokens per step?
- The reporting of extra parameters in Table 1 appears imprecise, as Medusa-2 requires tuning of the original LLM. Comparisons should focus on the number of tuned parameters rather than extra parameters.
- In line 403, the authors claim that "the generation quality of Speculative Streaming consistently outperforms that of next-token prediction-based fine-tuning, positioning it as a compelling alternative to LoRA-based fine-tuning approaches." However, no comparison with next-token prediction-based fine-tuning methods (such as LoRA tuning) is provided, which makes this an overstatement.

---

> ### Author Response · Authors · 2024-11-19
> **Addressing reviewer comments**
>
> We sincerely appreciate the insightful feedback and thoughtful inquiries provided by the reviewer. In this response, we aim to comprehensively address the concerns and questions raised, hoping to clarify and enhance the understanding of our research.
>
> **The original LLM distribution is altered**
>
> Thank you for your thoughtful feedback. We acknowledge the importance of preserving the original LLM distribution in speculative decoding frameworks. As detailed in Section 3.1.4, to address different application scenarios, Speculative Streaming (SS) supports both "lossy" mode a.k.a. share mode (that you are referring to), and "lossless" mode (that you might not have noticed) and preserves output distributions similar to [1, 2, 3] . More specifically:
>
> * In the **lossy (shared) mode**, parameter sharing occurs between LoRA adapters for next-token prediction and speculative draft generation. This design minimizes parameter overhead on constrained devices and is supported by observed gains in metric performance with shared parameters. Although, as you mentioned, this mode does not guarantee the preservation of the output distribution, it still outperforms the baseline, as demonstrated in Table 1.
> * In the **lossless mode**, speculative adapters operate independently of the base adapters and only speculative adapters in MSA layers are trained. Similar to [1, 2, 3], this ensures the model’s output distribution remains identical to that of the original LLM, preserving generation quality. Details of this approach are provided in Section 3.1.4 and the Metrics subsection of the Experiments section.
>
> For generic instruction-tuning tasks, where the base model is already well-trained, the lossless mode is employed, as highlighted in the Metrics subsection. This approach ensures the integrity of the output distribution and addresses concerns about sensitivity to fine-tuning and evaluation domains, as evidenced by the results in Figure 3, 4.
>
> **Lack of Experimental Detail**
>
> Thank you for raising this point. As noted in Section 3.1.4, Speculative Streaming (SS) supports both lossy (shared) and lossless modes. In the lossless mode, speculative adapters operate independently of base adapters, ensuring that the model’s output distribution matches the original LLM without requiring fine-tuning of base adapters. We only tune speculative adapter parameters on instruction following dataset - Vicuna as noted in section 3.1.4. The multi-stream mechanism and speculative candidate generation remains consistent across both modes, as detailed in Section 3.1.1. We are happy to include additional experimental details to clarify this setup further if needed.
>
> **Computation overhead**
> Thank you for your observation. While Speculative Streaming (SS) does slightly increase FLOPs overhead due to the additional stream representations, it remains substantially more efficient than LooKAhead decoding, as shown in Figure 6. Additionally, decoding is often memory bandwidth-bound on mainstream accelerators, meaning the slight increase in compute overhead does not lead to increased decoding latency, as demonstrated in Figures 3 and 4.
>
> **tree drafting strategy for Medusa**
>
> we use optimal configurations for speculative tree generation for both Medusa and Speculative streaming for fair comparison as noted in Appendix G.1.
>
> **Call Reduction Ratio Metic**
>  CR ratio denotes ratio between number of model.forward() calls required for auto-regressive decoding to those required for speculative-streaming.
>
> **Reporting of extra parameters in Table 1**
>
> Thank you for pointing this out. We agree that Medusa-2 also tunes the base LLM parameters, which highlights the parameter efficiency of our approach. We will update both tuned and extra parameters for Medusa-2 in the final revision to make this distinction.
>
> **Comparison with Lora Fine-tuning** :  In line 403, the authors claim that "the generation quality of Speculative Streaming consistently outperforms that of next-token prediction-based fine-tuning, positioning it as a compelling alternative to LoRA-based fine-tuning approaches." However, no comparison with next-token prediction-based fine-tuning methods (such as LoRA tuning) is provided, which makes this an overstatement.
>
> Thank you for the feedback. We would like to clarify that Table 1 provides a direct comparison between Speculative Streaming and next-token prediction-based fine-tuning (LoRA). As shown in the table 1, Speculative Streaming consistently achieves higher generation metrics compared to LoRA-based fine-tuning, supporting our claim.

---

> ### Comment · Reviewer_hmCo · 2024-11-26
>
> Thanks for the authors' prompt response. I still have the following concerns:
>
> 1. **Q1: The original LLM distribution is altered**. Thanks for your clarification. I acknowledged the chatbot-like settings that $\alpha_0$ is set to 0. However, as you stated in Equation (1) that "We enable the main stream to attend to speculative streams, allowing it to plan its residual transformations based on anticipated future residual states." In this way, the original attention is altered, leading to changes in the LLM output distributions.  How do you ensure that the target LLM's original output distribution remains unchanged given these architectural changes?
> 2. **Q2: Lack of Experimental Detail**. In the abstract, Section 3.1.4, and Metrics, the authors claim that "in **lossless** settings for generic chatbot applications that **do not necessitate supervised fine-tuning**." However, in your response, you claimed that "We only tune speculative adapter parameters on instruction following dataset - Vicuna as noted in section 3.1.4." This apparent contradiction suggests a potential **overclaim** in the manuscript. Could you clarify whether **fine-tuning is indeed required** for both lossy and lossless modes?
> 3. **Q3: tree drafting strategy**.  The referenced Appendix G.1 appears to be missing from the manuscript. There is only Appendix G and I could not find comparisons between Medusa and SS. Besides, please provide the average draft length after pruning.
> 4. **Q4: the CR (call reduction) ratio metric**.  You define the CR (call reduction) ratio as"the ratio between number of model.forward() calls required for auto-regressive decoding to those required for speculative-streaming." Is it equal to the mean number of generated tokens per step? Why don't you use this metric for consistency with prior work?
> 5. **Q5: Comparison with Lora Fine-tuning.** Thanks for the clarification that "Table 1 provides a direct comparison between Speculative Streaming and next-token prediction-based fine-tuning (LoRA)." Do you mean that the baseline is trained by LoRA? I noticed that the "# Extra Parameters" is set to 0 for the baseline. Do you mean that LoRA parameters are not counted as extra parameters in SS?

---

> > ### Author Response · Authors · 2024-11-27
> >
> > **Q1: The original LLM distribution is altered**
> >
> > Thank you for bringing up this important concern. We appreciate the opportunity to elaborate further. In **shared mode**, Speculative Streaming enhances generation metrics on downstream tasks by allowing the main stream to attend to anticipated future residual states (Equation 1). However, in **lossless mode**, the main stream does not interact with speculative streams—it neither attends to them nor shares trainable parameters with speculative stream adapters. This ensures that the original residual transformations of the base model, and consequently its output distribution, remain unaltered. To address related questions you had previously raised, we have added a dedicated section (Section 3.1.5) in the revised manuscript to clearly explain the distinctions between shared and lossless operating modes.
> >
> > **Q2: Lack of Experimental Detail**
> >
> > Thank you for raising this important point. To clarify, our statement refers to the fact that for generic chatbot tasks, there is no need to fine-tune speculative adapters on task-specific datasets that vary based on the task. Instead, we fine-tune speculative adapters once on an instruction-following dataset, Vicuna. Once trained, these adapters can be applied to generic tasks such as those in the MT-Bench dataset (e.g., reasoning, coding, writing).  For application-specific scenarios where the base LLM requires fine-tuning for specific applications (e.g., SQL generation), we share the adapter parameters between the main and speculative streams and fine-tune them jointly.
> >
> > We hope this clarifies the distinction. We have also revised the phrasing in the updated manuscript (abstract, Section 3.1.4, and Metrics ) to avoid any confusion and added section 3.1.5 for clear distinction between shared and lossless modes . Please let us know if additional details are needed.
> >
> > **Q3: tree drafting strategy**
> >
> > Thank you for pointing out the missing reference. We apologize for the incorrect appendix section number. After updating the headers, the ablation section was moved to the appendix, causing all appendix sections to shift by one. The correct reference is Section H.1.
> >
> > For speculative tree draft generation, we used the optimal settings for both Medusa and Speculative Streaming, as outlined in Appendix H.1, specifically `gamma = 3` and treeFactor  `k = 4`. We pass 32 nodes as a tree draft for speculative streaming after the pruning layer while in case of Medusa we pass 64 nodes, as these configurations yield the best wall-time speedups for respective approaches.  We added these details in appendix H.1 of updated manuscript. Please let us know if you need further clarification.
> >
> > **Q4: the CR (call reduction) ratio metric**
> >
> > Thank you for the suggestion. Our focus in this work was on minimizing auto-regressive calls due to the significant latency and power overhead associated with each call, particularly on edge devices. This is why we used the "call reduction" terminology. However, we acknowledge that it is equivalent to the mean number of generated tokens per step. We have updated the terminology in the revised manuscript for clarity and consistency.
> >
> > **Q5: Comparison with Lora Fine-tuning**
> >
> > Yes, your understanding is correct. In Table 1, the baseline refers to the base LLM fine-tuned on a specific application using LoRA, and the LoRA parameter overhead is considered as 0. For each of the target approaches, we list the number of extra trainable parameters required (in addition to the base LoRA).
> > In **shared mode**, since Speculative Streaming shares parameters with the base adapters, we do not count these as additional parameters. The overhead for Speculative Streaming mainly consists of stream embeddings and the tree pruning adapter (Table 1).
> > In **lossless mode**, since there is no parameter sharing, we count all additional speculative adapter parameters, along with stream embeddings and the tree pruning adapter, as part of the parameter overhead for Speculative Streaming (Figure 5).
> >
> > We hope this addresses your concerns. Please let us know if you have any further questions, comments, or concerns. Thank you again for your valuable feedback.

---

> > > ### Comment · Reviewer_hmCo · 2024-12-01
> > >
> > > I appreciate the author's feedback and modifications on the revised manuscript, which have provided important clarifications regarding the main contributions of their work and mitigated potential misunderstandings of readers. However, I maintain my opinion that the paper needs further modifications for the next submission. Several concerns are summarized below:
> > >
> > > 1. **W1: Architectural Complexity:** Unlike previous approaches (e.g., Medusa and Eagle) that implement a single lightweight module, SS nessisitates the fine-tuning of both LoRA adapters and stream embeddings, which increases the computational complexity and deployment overhead of the LLM.
> > > 2. **W2: Misleading of the main figure:** As the authors stated, LoRA tuning is required in both lossless and lossy modes of SS. However, this part of parameters is **omitted** in the main Figure (Figure 2), which could mislead readers into assuming that stream embeddings are the only parameters that requires tuning.
> > > 3. **W3: Computational overhead:** While SS may require fewer additional parameters compared to Medusa/Eagle, its stream representations necessitate forward passes through the final $N_s$ MSA layers, including both MLP and attention components. This adds an additional computational burden to the whole framework (more than Medusa as shown in Figure 6). This limitation warrants more thorough discussion and should be reflected in Tables 1 and 2.
> > > 4. **W4: Extra parameters:** SS nessistates fine-tuning of both LoRA and stream embeddings, all parameters tuned should be included as extra parameters in Table 1 \& Table 2. Besides, the reporting of extra parameters in Table 1 appears imprecise, as Medusa-2 requires tuning of the original LLM. Comparisons should focus on the number of tuned parameters rather than extra parameters. This part is still not fixed in the revision.
> > > 5. **W5: Draft Call Comparison:** The comparison of draft calls is not appropriate in Table 2. Though SS do not requires multi-round draft calls, it requires additional training while two-model SD is a plug-and-play method.
> > > 6. **W6: Overclaim:** In Line 421-425, the authors claim that "In lossless settings, our method consistently achieves better speedup than alternative approaches across Vicuna and Llama models of various scales (see Figure 3 and Figure 4), while incurring **significantly lower memory access and computational overhead** (see Figure 5 and Figure 6)." However, as pointed in W3, SS requires more computational overhead than Medusa and Eagle, which seems to be an **overclaim**. I appreciate that the authors has fixed the overclaim of "in lossless settings for generic chatbot applications, SS **does not necessitate supervised fine-tuning**." in the revision. However, over claims still exsit in the manuscript, which impacts the overall evaluation of the work. The wording should be carefully examined and organized in the next submission.
> > >
> > > I appreciate the authors' dedication to improving the manuscript through this review process. I maintain my current evaluation score based on these considerations.

---

> > > > ### Author Response · Authors · 2024-12-03
> > > >
> > > > **Deployment overhead**
> > > >
> > > > We respectfully disagree with this assessment. SS significantly reduces deployment overhead as the combined number of trainable parameters, including stream embeddings and LoRA adapters, is substantially lower than those of Medusa and Eagle, as shown in Table 1. Furthermore, stream embeddings and LoRA adapters are not standalone modules; they are integrated into a unified training procedure, which can be executed either jointly with the base model or independently, similar to Medusa and Eagle.
> > > >
> > > > **Misleading of the main figure**
> > > >
> > > > We appreciate this observation and would like to clarify that an update symbol in the top-left corner of Figure 2 indicates that LoRA adapters are trained. This is explicitly discussed in multiple sections of the manuscript (e.g., Sections 3.1.5 and 4).
> > > >
> > > > The primary purpose of Figure 2 is to provide a high-level architectural overview of SS during both training and inference. Detailed training procedures, including LoRA tuning, are elaborated upon in a dedicated section to avoid overloading the figure with specifics. Nonetheless, we will revise the caption of Figure 2 to further emphasize that both stream embeddings and LoRA adapters are tuned, ensuring there is no ambiguity for the reader.
> > > >
> > > >
> > > > **Computational overhead**
> > > > We acknowledge that SS incurs a modest increase in FLOPs relative to Medusa and Eagle. However, modern GPUs and mobile neural engines are optimized for high computational throughput and typically possess orders of magnitude more available FLOPs compared to their memory bandwidth. The **primary goal of SS is to leverage this surplus computational capacity to reduce overall decoding latency** by increasing the arithmetic intensity of the decoding process. We did not include these details in Tables 1 and 2 as they are already thoroughly addressed in Figure 6. Furthermore, the discussion on arithmetic intensity and its role in optimizing inference is elaborated in Appendix F.
> > > >
> > > >
> > > > **Extra parameters**
> > > > We appreciate the reviewer's feedback and respectfully clarify our approach. In the supervised fine-tuning setup, the base model is fine-tuned using LoRA adapters, regardless of the speculative decoding approach (SS, Medusa, etc.). In Table 1, we specifically report the extra parameters required in addition to the base LoRA adapters. For lossless settings, where the base model remains frozen, all tunable parameters—including LoRA adapters and stream embeddings—are accounted for as overhead in Figure 5. This separation ensures consistency in reporting across different setups while maintaining comparability between methods.
> > > >
> > > >
> > > > **Draft Call Comparison**
> > > >
> > > > We respectfully disagree with the characterization of the two-model approach as a plug-and-play method. In application-specific settings (e.g., SQL generation), the draft model must be aligned with the target model to achieve high acceptance rates, as discussed in Section 1. This alignment necessitates fine-tuning similar to SS. In all our experiments, we fine-tuned the draft model either in a supervised fine-tuning setup or on an instruction-tuning dataset to ensure a fair comparison with SS.
> > > >
> > > > The purpose of Table 2 is to compare the overhead of autoregressively generating speculation (as required by two-model SD) with SS, which generates speculation in a non-autoregressive manner. By leveraging the memory bandwidth bottlenecks of modern accelerators, SS achieves significant computational efficiency.
> > > >
> > > > **Overclaim**
> > > >
> > > > We respectfully maintain that our claim is not an overstatement. SS significantly reduces memory access overhead compared to all competitive approaches, as demonstrated in Figure 5. Additionally, while SS incurs slightly higher FLOP overhead than Medusa and Eagle, it remains substantially lower than approaches like lookahead decoding, achieving an effective balance between memory access and computational efficiency. This trade-off positions SS as a practical middle ground, leveraging available computational resources in memory bandwidth bound decoding settings. We will revise the wording in the final revision to ensure clarity and address the reviewer's concerns about potential overclaims.

---

### Official Review · Reviewer_JpUo · 2024-10-30

**Soundness:** 4
**Presentation:** 3
**Contribution:** 3
**Rating:** 8
**Confidence:** 4

**Summary:**

This paper introduces Speculative Streaming, a novel approach to make LLM inference faster without needing extra models. It combine speculation and checking in one model with multi-stream attention and put future token planning in fine-tuning. Speculative Streaming make inference 1.9-3X faster for different tasks, make generation better, and use ~10000X less extra parameters than other methods. It good for devices with not much resource and can be used for chatbots without losing quality.

**Strengths:**

1. The novelty is good. This paper presents an innovative approach to speculative decoding that eliminates the need for auxiliary draft models, addressing a significant limitation in existing methods.
2. The method is efficient. Speculative Streaming demonstrates impressive speedups (1.9-3X) while using substantially fewer parameters than competing approaches, making it highly efficient and suitable for resource-constrained environments. The speedup is consistent  across diverse tasks, including summarization, structured queries, and meaning representation, indicating its broad applicability.
3. The method works well for both fine-tuned and no-loss settings, so it useful for many applications, from specific tasks to general chatbots.

**Weaknesses:**

The method was tested only on model with a size < 7B parameters, a more detailed analysis of how it scales with very large models (e.g., 100B+ parameters) would be valuable. And a more in-depth theoretical analysis of why Speculative Streaming works would be helpful, particularly in terms of the interplay between multi-stream attention and future token planning.

**Questions:**

1. How does Speculative Streaming perform on very long sequences (e.g., 10,000+ tokens)? Does the efficiency gain remain consistent?
2. Can you provide more insights into how the multi-stream attention mechanism affects the model's understanding of context and coherence in generated text?
3. How sensitive is the method to hyperparameters, particularly the number of speculative streams? Is there an optimal range for different model sizes or tasks?

---

> ### Author Response · Authors · 2024-11-19
>
> **Speculative Streaming performance on very long sequences**
>
> To evaluate the performance of Speculative Streaming on long sequences, we trained speculative adapters on the Arxiv-summarization dataset and tested it on the Summarization task from the LongBench dataset. Since the KV cache is shared between the main and speculative streams, there is no additional runtime memory overhead. While compute in attention layers increases due to longer context, the compute in MLP layers remains the same, and decoding is still memory bandwidth bound. We achieved a 2.64X speedup on the Summarization test set using gamma = 3 and treeFactor = 4. We include these experiments in Appendix K of the revised version.
>
> **Effect of multi-stream attention mechanism on the model's understanding of context and coherence**
>
> The multi-stream attention mechanism improves context understanding and coherence by enabling the primary stream $M_t$ to incorporate information from speculative streams $S_{tj}$, which approximate future residual states. This mechanism affects the model in three key ways:
>
> 1. **Improved Predictive Transformations**: By attending to $S_{tj}$,  $M_t$ refines its residual transformations, incorporating predictive signals about token relationships over the next $\gamma$ tokens. This enhances anticipatory planning during training, leading to more coherent token transitions.
>
> 2. **Dynamic Context Alignment**: Speculative streams progressively align with future residual states, helping $M_t$ better encode dependencies between current and future tokens. This reduces mismatches in long-range coherence.
>
> 3. **Enhanced Representation Fidelity**: Multi-stream attention ensures that $M_t$ captures a richer, temporally-aware context, leading to representations that align closely with the actual ground truth dynamics, thus improving generation quality metrics while maintaining coherence.
>
> **Sensitivity of the method to Number of streams and hyperparameters**
>
> We observe diminishing returns in speculative prediction accuracy with an increasing number of speculative streams. As detailed in the following table, stream prediction accuracy decreases by approximately 5-10% with each additional stream:
>
> | Stream Number | Accuracy (%) |
> |---------------|--------------|
> | Stream 1      | 71.12        |
> | Stream 2      | 65.43        |
> | Stream 3      | 56.09        |
>
> It’s important to note that with each added stream, the size of the speculative window grows quadratically, as discussed in Section 3.1.3. Based on this trade-off, we find that using 3 streams is near optimal for most tasks.
>
> Regarding other hyperparameters, extensive experiments indicate that the following setup yields robust performance, achieving substantial speedups across different tasks and model sizes:
>
> - **Number of streams**: 3
> - **Number of MSA layers**: 1/8 of base model layers
> - **LoRA adapter rank**: 8
> - **Tree Factor**: 4
> - **$\alpha_0$**: 1
> - **$\alpha_1, \alpha_2, \alpha_3$**: 0.1

---

> > ### Comment · Reviewer_JpUo · 2024-11-26
> > **Thanks for the clarification.**
> >
> > I don't have further questions.

---

### Official Review · Reviewer_1U6S · 2024-11-02

**Soundness:** 3
**Presentation:** 2
**Contribution:** 3
**Rating:** 6
**Confidence:** 4

**Summary:**

This paper presents speculative streaming, a parameter-efficient method for speculative decoding. Speculative streaming leverages multi-stream attention to integrate representations of future token predictions directly into the language modeling process. Unlike traditional approaches, it uses a multi-token prediction strategy during training rather than the standard next-token prediction. In evaluations, the proposed method demonstrated superior performance in both latency and generation quality compared to several baseline methods.

**Strengths:**

**Parameter Efficiency**: The proposed multi-stream attention method effectively reduces the parameter requirements for speculative decoding.


**GPU Utilization**: By design, the multi-stream attention architecture achieves higher GPU kernel utilization, providing a notable advantage over popular methods like Medusa.


**Evaluation Result**: Speculative streaming requires significantly fewer parameters than other speculative decoding approaches, enhancing both efficiency and practicality.

**Weaknesses:**

**Model Evaluation**: Since the proposed training method modifies the original model output, a thorough evaluation is essential to verify the claimed integrity of the target model. While model performance is reported on the SqlContext, Dialogsum, and e2e-nlg datasets using metrics like exact match and ROUGE score, these evaluations may not fully capture the performance of a modern LLM, such as LLaMA.

**Formatting Issues**: The citation format does not meet ICLR requirements, and Figure 17 appears distorted. The titles in the headers are not rendered.


**Lack of Throughput Experiments**: The experiment is conducted on an Nvidia A100 GPU with a batch size of 1, and an ablation study with batch sizes of 2 and 4 is included in the appendix. However, using an A100 for inference with a batch size of 1 is uncommon in practice. An evaluation of throughput (at maximum batch size) compared to other methods is necessary to demonstrate the practical performance of speculative streaming.


**Lack of Motivation**: The paper emphasizes parameter efficiency in the speculative process, though a draft model may contain as little as 1% of the parameters of the target model. The significance of this efficiency focus is not shown and would benefit from further justification.

**Questions:**

1. In Figure 3 and Figure 4, do you achieve a 3-time speedup in comparison to 2-model SD?
2. What prompts are used during the evaluations?
3. Are the target model weights frozen in all experiments? Why do you choose to “fine-tuning only LoRA parameters of MSA layers”?
4. You mentioned “Furthermore, it is important to highlight that the generation quality of Speculative Streaming consistently outperforms that of next-token prediction-based finetuning, positioning it as a compelling alternative to LoRA-based fine-tuning approaches.” in line 405.” What is the evidence of this?

---

> ### Author Response · Authors · 2024-11-19
>
> We sincerely appreciate the insightful feedback and thoughtful inquiries provided by the reviewer. In this response, we aim to comprehensively address the concerns and questions raised, hoping to clarify and enhance the understanding of our research.
>
>
> **Model Evaluation**
> Thank you for your comment. Speculative Streaming operates in both **lossy(shared)** and **lossless** modes, as described in Section 3.1.4. In lossless mode, speculative adapters operate independently of the base adapters and the model output distribution remains identical to the base model. We provide extensive evaluation in Figures 3 and 4, showcasing speedups on Vicuna-7B, Vicuna-13B, LLaMA-7B, and LLaMA-13B, comparing our approach with state-of-the-art speculative decoding methods.
> We employ “lossy” (shared) mode in application specific settings, where parameters are shared between the LoRA adapters for next-token prediction and speculative draft generation to minimize the trainable parameter overhead on resource-constrained devices. In application specific settings we report metrics recommended for those tasks while in generic chatbot settings, we keep the output distribution same and only report speedup as noted inn Metrics subsection.
>
> **draft model may contain as little as 1% of the parameters of the target model**
>
> Thank you for the insightful comment. As highlighted in Section 1, in application-specific settings, a dedicated draft model must be trained to align with the fine-tuned target model for optimal acceptance rates. As the number of applications increases, the **cumulative parameter overhead** can become significant. The memory resources required to host these application-specific draft models could instead be allocated to training a larger and more capable target model.
>
> While parameter efficiency is an important advantage of our approach, it is not the sole benefit. As demonstrated in Table 2, using a draft model that is 10-100x smaller in traditional speculative decoding results in suboptimal speedups. Appendix G provides a detailed analysis of the model size versus speedup trade-off in two-model speculative decoding, showing that excessively small models compromise acceptance rates, while larger draft models increase generation time, limiting overall speedup. In contrast, Speculative Streaming leverages the full LLM, eliminating the trade-offs between draft generation and draft acceptance. Because speculative tokens are generated within the same forward pass, there is no additional latency, and the decoding process remains memory bandwidth-bound on most mainstream accelerators. This ensures both high acceptance rates and substantial speedups without compromising efficiency or performance.
>
> **Formatting Issues** Thanks for pointing out the compile issues. We have fixed all formatting issues in the revised submission.
>
> **speedup in comparison to 2-model SD?**
> Yes your understanding is correct. Our method achieves 3X speedups relative to 2-Model SD on Vicuna-7B, Vicuna-13B, Llama-7B, LLama-13B as shown in Figure3 and 4.
>
> **What prompts are used during the evaluations?**
>
> Thank you for your question. For LLM evaluations, we use the prompts from the [FastChat repository](https://github.com/lm-sys/FastChat). As mentioned in the Metrics subsection, for instruction following tasks, our method trains only the speculative adapters, ensuring that the model’s output distribution remains identical to that of the baseline model.
> In supervised fine-tuning setups, such as for SQL response generation, we compute the exact match between the generated and target responses using the Huggingface library.
>
> **Are the target model weights frozen in all experiments?**
>
> Yes, the target model weights are frozen in all experiments. In Application specific settings, we use lossy(shared) mode and share LoRa parameters between main and speculative streams and train them jointly. In “lossless” mode, we use separate speculative adapters with no parameter sharing, ensuring the model’s output distribution remains identical to the base model, as noted in Section 3.1.4 and the Metrics subsection.
>
>
> **Evidence of generation quality**
>
> In Table 1, we provide a direct comparison of generation metrics between next-token prediction-based fine-tuning (LoRA) and Speculative Streaming. Speculative Streaming consistently achieves superior generation metrics.
>
> **Experiments with higher batch sizes**
>
> Thank you for your feedback. Speculative Streaming is primarily designed for resource-constrained edge device use cases where batch size is typically 1. However, as noted in the ablation section, we also explore the use of larger batch sizes (2 and 4) for server-based models, with minimal impact on generation speedups. We will consider including a more thorough comparison of throughput at higher batch sizes in final revision to demonstrate the practical performance of Speculative Streaming across different deployment scenarios.

---

> > ### Comment · Reviewer_1U6S · 2024-11-21
> >
> > Thank you for the clarification.
> > Could you clarify what is the baseline in Table 2 of the current version?
> > Regarding Figures 3 and 4, I would also like to see the comparison to autoregressive generation.

---

> > > ### Author Response · Authors · 2024-11-21
> > >
> > > Thank you for your thoughtful feedback and for pointing out areas where we can provide additional clarity.
> > >
> > > * The baseline approach presented in the table 2 of the current version corresponds to 2-model speculative decoding, using OPT-125M as the draft model and OPT-1.3B/OPT-6.7B as the target model. In contrast, the proposed approach, speculative streaming, operates directly on the target models (OPT-1.3B and OPT-6.7B).
> > > * In Figure 3, the speedup achieved by the autoregressive baseline is 0.86 on the same scale as other approaches for the Vicuna-7B model and 0.78 for the Vicuna-13B model.
> > >
> > > We hope this response clarifies and addresses your concerns, but we would be happy to answer any follow up questions you might have. We sincerely value your feedback and are committed to improving the clarity and quality of our work.

---

> > > > ### Comment · Reviewer_1U6S · 2024-11-22
> > > >
> > > > Thank you for the further clarification. Some of my concerns have been addressed, and I have updated my rating accordingly.
> > > >
> > > > However, I believe the experiments with 2-model speculative decoding are not properly conducted, as the draft model is not fine-tuned on ShareGPT like the other methods. This creates a specific disadvantage for the 2-model speculative decoding method. I hope the authors can update the experiment using a fine-tuned draft model for the 2-model speculative decoding to ensure the fairness of the comparison.

---

> > > > > ### Author Response · Authors · 2024-11-23
> > > > >
> > > > > Thank you again for participating in the discussion with us and thanks for increasing your score. Just for the clarification, we want to point out that the draft model used in the speculative decoding experiments presented in Table 2 is fine-tuned on each respective application, and for experiments in Figures 3 and 4, it is fine-tuned on the Vicuna dataset, consistent with the training protocol for other single-model baselines. Therefore, the two-model speculative decoding method does not suffer from a disadvantage in terms of training. We will make sure to clarify this further in the final version. We have also included a discussion on the 2-model speculative decoding speedup limitations in Appendix H highlighting that excessively small draft models reduce acceptance rates, while larger draft models increase speculative draft generation time, limiting overall speedup. We hope this addresses your concern.

---

> > > > > > ### Comment · Reviewer_1U6S · 2024-11-24
> > > > > >
> > > > > > I have updated my score, as all of my concerns have been addressed. I hope the author will consider incorporating my suggestions into the final version of the paper.

---

### Official Review · Reviewer_KiEv · 2024-11-04

**Soundness:** 2
**Presentation:** 3
**Contribution:** 2
**Rating:** 5
**Confidence:** 4

**Summary:**

This paper proposes a non-autoregressive draft embedding to accelerate speculative decoding.

**Strengths:**

This paper is generally well-written and easy to follow.

The experiments of this paper are detailed and the authors test different llms, which is good.

**Weaknesses:**

1) The method of this paper is very similar to BiTA, https://arxiv.org/html/2401.12522v2.

2) Due to the params of LLM are also updated, I just wondering if the training dataset will introduce bias. For example, if the training set is SQL, the metric of speculative streaming may be very good but fails at other tasks, e.g., machine translation. Can the authors add lossless acceleration result like eagle?

**Questions:**

see cons

---

> ### Author Response · Authors · 2024-11-19
>
> **The method of this paper is very similar to BiTA, https://arxiv.org/html/2401.12522v2.**
>
>
> Thanks for bringing this work to our attention. While there are similarities, such as using masked tokens for speculative generation, we highlight the following key differences and novelties in our approach:
>
>
> **Attention Mechanism with Speculative Residual States**
>
> Our approach features a novel attention mechanism where the main stream also attends to speculative residual states, in contrast to the BiTA paper, where speculative streams attend to the main stream without reciprocity. This design improves generation metrics by enabling next-token planning (Section 4.1.2).
>
> **Tree Pruning for Computational Efficiency**
>
>    We propose a Tree Pruning mechanism where we prune tree draft nodes that are less likely to be accepted during final verification to reduce both compute and memory overhead, further accelerating decoding (Section 3.1.3, Appendix A.1 (B.1 in new revision)). In contrast, BiTA has no such mechanism.
>
> **Trainable Parameters**
>
>    Unlike BiTA, our work focuses on NAR speculative draft generation using minimal trainable parameters (See Table 1) to transform residuals in each layer in form of speculative LoRA adapters, tailored for resource-constrained settings.
>
> **Shared vs. Lossless Modes for LoRA Adapters**
>
> Our work introduces shared and lossless modes for LoRA adapter parameters between the main and speculative streams, offering flexibility in balancing the number of trainable parameters and preserving the output distribution's integrity. We employ the shared mode for application-specific tasks and the lossless mode for general-purpose chat scenarios. In contrast, BiTa does not make this distinction.
>
>
> **Training Dataset Bias**
>
> We want to note that, as mentioned in the datasets subsection, we studied two settings: 1) Application-specific scenarios as you are referring to, where we perform lightweight supervised fine-tuning with LoRA adapters that are shared between main and speculative streams on a specific task (e.g., SQL), introducing minimal trainable parameter overhead (Table 1 and 2) and general instruction-following setting where adapters are not shared between main and speculative streams and only speculative adapters are trained on instruction set  similar to Eagle. As reported in Figure 3, without any task-specific fine-tuning of base model parameters, we achieve 3X speedup on MT Bench, which includes tasks across categories like writing, coding, and reasoning while maintaining integrity of base model’s output.

---

> > ### Comment · Reviewer_KiEv · 2024-11-20
> >
> > Thank you for the author's response. However, I do not believe it addresses my concerns; rather, it exacerbates them.
> >
> > **Attention Mechanism with Speculative Residual States**: In BITA, the LLM generation process is not affected by the draft LM, whereas the method proposed by the authors clearly introduces interference. I do not see this as an advantage.
> >
> > **Tree Pruning for Computational Efficiency**: Tree pruning has already been proposed by GLIDE with CAPE and Eagle-2. It is not a substantial contribution.
> >
> > **Trainable Parameters**: From my perspective, the number of trainable parameters in BITA is not significantly higher than in the method discussed in this paper.
> >
> > **Training Dataset Bias**: Unless the authors test on a broader dataset, such as SPEC-Bench, I cannot accept the results based on MT-Bench alone.
> >
> > If the authors cannot effectively address my concerns, I will lower the score to 3.

---

> > ### Comment · Reviewer_KiEv · 2024-11-20
> >
> > Does "lossless" in this paper mean "lossless" relative to the **original** LLM?

---

> ### Author Response · Authors · 2024-11-21
>
> Thank your thoughtful feedback and critique, and participating in the discussion.  Below, we address each of the concerns raised and aiming to clarify the contributions of our work.
>
> 1. **Acknowledgment of Concurrent Work**:
>
> We appreciate references to BiTA, Glide with CAPE, and Eagle-2.
>
> First, please note that **BiTA and “Glide with CAPE”** were appeared publicly within **20 days of our work**, and **Eagle-2** was appeared **4 months later** and we were unaware of them during the development of our work. That said, regardless of the timeline, we acknowledge the relevance of all these interesting works and clarify our technical differences below. We will also include a detailed discussion in the revised paper to acknowledge their contributions and to clearly delineate the differences and novelties of our approach.
>
> 2. **Comparison to BiTA**:
>     While we acknowledge similarities with BiTA, our approach has some key distinctions:
>
>     2.1 **Training-Inference Consistency**: BiTA’s Section 3.2 highlights a mismatch in attention, where mask tokens attend to all previous tokens during training but not during inference. As mentioned in section 3.2 mask tokens do **NOT** attend to draft tokens in speculative window. To illustrate this inconsistency, we quote two relevant sentences from Section 3.2:
>     - “Note that they do not “attend” to the current draft token candidates, generating future predictions in the context of only both query tokens and previously generated tokens. “
>     - “As stated, the draft token candidates are predicted based on the context of the output sequence until the last forward pass, rather than incorporating the currently accepted draft tokens.”
>
>    On the other hand, as demonstrated in Section 3.1.2, our method employs  γ streams per token during the verification step. Each of these  γ streams attends to both the previous streams and the preceding context, ensuring consistency between training and inference.
>
>    2.2 **Flexibility between shared and lossless modes**: Unlike BiTa, which enforces a strict separation between mask tokens and base model parameters to keep generation quality same, our approach in **shared mode** improves both generation quality and speedup in application specific settings. We find that, generation quality of responses  can be improved significantly by having main stream attend to speculative streams that resemble residual states of future tokens leading to better next token planing (see section 4.1.2) and subsequently better generation quality as demonstrated in Table 1.  On the other hand, we also offer a **lossless mode** in generic chatbot settings that is guaranteed to keep output distribution same as base model.
>
>
>    2.3 **Memory and Compute Efficiency**: BiTA applies Masked attention across all layers, whereas we restrict the use of multi-stream attention to Ns=4 layers (Sections 3.1.2 and 3.1.3). This design reduces forward-pass latency and FLOP overhead, particularly as model transitions into compute-bound regimes with larger tree sizes.
>
> We believe these design differences contribute to our higher speedup compared to BiTA (3.0x vs. 2.38x on LLaMA-7B as measured on MT-Bench).
>
> 3. **Tree Pruning for Computational Efficiency**:
>
>     * While Glide with CAPE and Eagle-2 use confidence-based pruning directly on speculative draft logits, our approach introduces a separate filter for speculative token pruning. Speculative tokens generated in a non-autoregressive (NAR) manner are pruned for **dependency enforcement using the initial layers of the target model via early exiting** in next forward pass. As illustrated in Figure 11, we rely on early-exited logits for pruning rather than those produced by the NAR pass.  By introducing dependency among consecutive nodes in speculative tree, we improve pruning efficacy leading to better speedups (See Figure 9).
>
> 4. **Trainable Parameters**:
>
>     * We respectfully clarify that our method introduces fewer trainable parameters compared to BiTA. Specifically, while BiTA adds soft prompt tokens across all layers, we limit speculative adapter training to \( N_s = 4 \) layers while still surpassing BiTa in terms of speedup on MT Bench (3.0x vs. 2.38x on LLaMA-7B).
>
> 5. **Dataset Scope**:
>     We appreciate your suggestion to evaluate on SPEC-Bench. We are actively extending our experiments to include broader datasets and will incorporate these results in the final version to further substantiate our claims.
>
> 6. **lossless relative to the original LLM?**
> Yes, that’s correct. It is lossless relative to the original LLM and  guarantees no change to the output distribution of the original model.
>
> We hope this response clarifies and addresses your concerns, but we would be happy to answer any follow up questions you might have. We sincerely value your feedback and are committed to improving the clarity and quality of our work.

---

> > ### Comment · Reviewer_KiEv · 2024-11-24
> >
> > Why can your method exceed bita on the lossless setting?

---

> > > ### Author Response · Authors · 2024-11-24
> > >
> > > Thank you again for participating in the discussion with us. We hypothesize that the reasons behind our method exceeding the speedup of BiTa on the MT benchmark in the lossless setting are as follows:
> > >
> > > -  **Training-Inference Consistency**: BiTA’s Section 3.2 highlights a mismatch in attention, where mask tokens attend to all previous tokens during training but not during inference. As mentioned in section 3.2 mask tokens do NOT attend to draft tokens in speculative window. On the other hand, as demonstrated in Section 3.1.2, our method employs γ streams per token during the verification step. Each of these γ streams attends to both the previous streams and the preceding context, ensuring consistency between training and inference, thereby enhancing ability to generate high quality speculative drafts.
> > >
> > > - **Dependency check for High Quality Drafts**:  As illustrated in Figure 11, our tree pruning mechanism ensures the elimination of low-quality speculative drafts where token dependencies are violated. For example, in a draft path like (“he,” “goes,” “went,” “to”) generated by NAR speculation, the path is automatically pruned because the early-exiting mechanism via virtual autoregressive model identifies that “goes” and “went” are not contextually dependent. This dependency check occurs in the **same forward pass during verification via early exiting**, without requiring additional autoregressive steps.
> > > Consequently, our approach passes higher-quality drafts with a greater likelihood of acceptance compared to BiTa, which only verifies NAR drafts generated by mask tokens in subsequent steps.
> > >
> > > - **Efficient Multi-stream Attention** : Our approach leverages only Ns=4 multi-stream attention layers as opposed to all layers used by BiTa, which reduces computation per forward step compared to BiTa. This computational efficiency enables us to utilize larger speculative tree sizes before transitioning into a compute-bound phase.
> > >
> > >
> > > We hope this clarifies our findings and the underlying advantages of our method. Thank you again for your valuable feedback.

---

### Meta-Review · Area_Chair_AKxA · 2024-12-18

**Metareview:**

This paper introduces Speculative Streaming as an attempt to unify speculation and verification in a speculative decoding setting, all in the same base model, without the need for an external auxiliary draft model. The proposal leverages what authors refer to as multi-stream attention, which is added to the base model, and enables that model to generate drafts beyond the next token. In doing so, forward passes can simultaneously be used to verify tokens generated so far, and speculate future tokens.

The main strength of this manuscript is how simple and potentially efficient the proposal is, with evaluations on different base models showing performances that match or improve upon recent methods such as Medusa, Hydra, and Eagle.

On the other hand, the presentation is not clear enough. The method description is a bit convoluted and contains more motivation than actual description of the method. The manuscript lacks a pseudo-code or something else that simply defines the inference procedure. Fine-tuning is also unclear and it's difficult to get from the manuscript what needs training.

The experiments are also such that it's non-trivial to draw conclusions from them. The sets of baselines change from evaluation to evaluation, without a clear justification. For example, for wall-clock time speedup comparisons, baselines in Figures 3 and 4 (and Figures 5 and 6) change. Details on fine-tuning are also missing. That is, do authors fine-tune on a composition of datasets or they fine-tune for each target task independently? Moreover, it's also unclear to what extent the evaluation is confounded by the implementations of baselines. The baseline results reported throughout the paper seem to correspond to the authors own implementations of these baselines. Ideally, we would additionally be able to compare the proposal's results against baseline performances obtained independently. Finally, authors center  the claims of improved efficiency on parameter efficiency, but gains in terms of FLOPs are not very pronounced. I would perhaps suggest a comparison in terms of inference throughput in tokens/s alongside memory footprint and task performance.

All in all, this paper requires improvements prior to publication for overall clarity and readability, but also for stronger evidence of the benefits of the proposal, which are a bit unclear in the current form.

**Additional Comments On Reviewer Discussion:**

This manuscript was a bit divisive among reviewers, but overall, concerns revolved around clarity, with concerns remaining for some reviewers even after multiple rounds of interaction. I would claim this as evidence that some improvement on the presentation side needs to be carried out in this manuscript.

---

### Decision · Program_Chairs · 2025-01-22

Reject